



# Influence of simple terrain on the spatial variability of a low-level jet and wind farm performance in the AWAKEN field campaign

William Radünz[1], Bruno Carmo[1], Julie K. Lundquist[2,3], Stefano Letizia[2], Aliza Abraham[2], Adam S. Wise[4,5], Miguel Sanchez Gomez[2], Nicholas Hamilton[2], Raj K. Rai[6], and Pedro S. Peixoto[7]

[1]Dynamics and Fluids Research Group, Escola Politécnica, University of São Paulo, São Paulo, SP, Brazil
[2]National Renewable Energy Laboratory, Boulder, CO, USA
[3]Johns Hopkins University, Baltimore, MD, USA
[4]University of California, Berkeley, CA, USA
[5]Lawrence Livermore National Laboratory, Livermore, CA, USA
[6]Pacific Northwest National Laboratory, Richland, WA, USA
[7]Institute of Mathematics and Statistics, Department of Applied Mathematics, University of São Paulo, São Paulo, Brazil

**Correspondence:** William Radünz (wradunz@usp.br)

**Abstract.** In wind energy research, scientific challenges are often associated with *complex terrain* sites, where orography, vegetation, and buildings disrupt flow uniformity. However, even sites characterized as *simple terrain* can exhibit significant spatial variability in wind speed, particularly during stable boundary layers (SBLs) and low-level jets (LLJs). This study investigates these terrain interactions using both simulations and observations from the American WAKe ExperimeNt (AWAKEN). We employ a multiscale Weather Research and Forecasting (WRF) model simulation, integrating mesoscale forcing in the coarse domains and representing three rows of turbines from the King Plains wind farm as generalized actuator disks (GAD) in the large-eddy simulation (LES) domains. During a nocturnal LLJ event on 3 April 2023, the downwind, wake-affected turbine rows outperformed the upwind, unwaked row by 25–51 %. This counterintuitive result arises from terrain-induced streamwise variations in hub-height wind speed of approximately 4 m s$^{-1}$ over 5 km—equivalent to ∼50 % of the upwind reference speed. This enhancement outweighs the wake-induced reduction in mean wind speed (∼12 %) and global blockage effects reported in the literature (∼1–3.4 %). The multiscale simulations capture the intra-farm spatial variability in power performance observed in SCADA data. Terrain-induced vertical displacement of the LLJ, coupled with large wind shear below the jet maximum, drives the substantial streamwise acceleration within the wind farm. These findings underscore the importance of accounting for spatial variability related to terrain, even in simple landscapes, particularly during LLJ conditions. Incorporating such effects into reduced-order modeling frameworks for wind farm design and control could significantly enhance their effectiveness.

*Copyright statement.* This work was authored in part by the National Renewable Energy Laboratory, operated by Alliance for Sustainable Energy, LLC, for the U.S. Department of Energy (DOE) under contract no. DE-AC36-08GO28308. Funding was provided by the U.S. Department of Energy Office of Energy Efficiency and Renewable Energy Wind Energy Technologies Office. The views expressed in the article do not necessarily represent the views of the DOE or the U.S. Government. The U.S. Government retains and the publisher, by





accepting the article for publication, acknowledges that the U.S. Government retains a nonexclusive, paid-up, irrevocable, worldwide license
22    to publish or reproduce the published form of this work, or allow others to do so, for U.S. Government purposes.

# 1   Introduction

24    One of the grand challenges in modern wind energy science is to measure, model and understand physical processes in
atmosphere–wind farm interactions that span a wide range of spatio-temporal scales (Veers et al., 2019, 2022). The soci-
etal motivation is to reduce the cost of energy, which can be achieved by at least two means. First, during the wind farm design
stage, an accurate understanding of the processes that modulate wind farm performance, backed up by observations and mod-
els, can help reduce uncertainty in the estimation of the annual energy production and design a wind farm that maximizes the
energy conversion. Second, for existing wind farms, appropriate observations and models that represent the relevant physical
processes are necessary for wind farm control. The existence of processes that create spatial gradients within the wind farm
area enhance the complexities associated with the design and optimization stages.

Spatial gradients in wind speed have paramount importance for the atmospheric sciences and wind energy. The vertical
gradient in wind speed is the wind shear and affects turbine performance (Sanchez Gomez and Lundquist, 2020; Murphy et al.,
2020) and loads (Sathe et al., 2013; Robertson et al., 2019; Lundquist, 2021). The wind results from interactions between
pressure gradient forces, Coriolis forces, and turbulent stresses. Horizontal gradients appear whenever a local imbalance in the
forces that drive the wind occurs, such as in the transition from surfaces of different roughness or heat flux (Stull, 1988), or for
flows over terrain with variable height (Baines, 1995). Horizontal gradients of velocity can be negligible compared to vertical
gradients over isothermal flat terrain with uniform roughness and thermal properties in the microscale range. Indeed, landmark
field campaigns were held in sites with flat and homogeneously covered terrain (Kaimal and Wyngaard, 1990; Holtslag et al.,
2012) to minimize such spatial variability.

During the nighttime over land, surface cooling produces thermal stratification that inhibits turbulent motions in the planetary
boundary layer, resulting in a stable boundary layer (SBL). A myriad of processes can occur in SBLs, such as atmospheric
gravity waves, topographic acceleration or blocking of winds, turbulence intermittency, and low-level jets (LLJs) (Poulos et al.,
2002; Fernando et al., 2019). A LLJ is a stream of fast-moving air with a maximum in wind speed relatively close to the ground,
which is enabled by the suppression of frictional forces in the upper portion of the SBL. Distinct physical mechanisms can form
LLJs (Blackadar, 1957; Holton, 1967; Stensrud, 1996; Banta et al., 2002; Banta, 2008; Klein et al., 2015; Smith et al., 2019a),
and once formed, it can manifest at least three turbulence regimes (Banta, 2008): a weakly-stable regime with continuous
turbulence, a very-stable regime with almost no turbulence, and a transitional or intermittent regime with occasional bursts of
turbulence. In the Southern Great Plains (SGP) of the United States, LLJs display these different turbulence regimes (Klein
et al., 2015) and accompanying large spatio-temporal variability in their evolution (Banta et al., 2002; Smith et al., 2019a).

There is an important link between terrain complexity, SBLs and LLJs in the modulation of the spatial variability of the
wind. The existence of horizontal variability introduces an additional complexity, because one can no longer assume a single
wind profile is representative of the entire site. In stable conditions, the low-level wind decelerates upstream of obstacles more





than it would in neutral conditions because of the downward buoyancy (Mahrt and Larsen, 1990; Baines, 1995; Hunt et al., 1988) forcing the flow to stay at the same altitude rather than rising over the obstacles. Likewise, the flow accelerates more in the lee, and this combination enhances the spatial variability in wind speed. This behavior happens, to a lesser or higher degree, to any SBL, from which the LLJs are a particular case. This variability is one of the main scientific challenges associated with complex terrain, and motivated for example a large-scale field campaign and model development effort in the New European Wind Atlas (NEWA) project (Mann et al., 2017). The Perdigão (Fernando et al., 2019) and the Alaiz (Santos et al., 2020) experiments from the NEWA revealed with unprecedented detail the variability in flow patterns that occurred when stable boundary layers (SBLs) and nocturnal low-level jets (LLJs) interacted with complex terrains (Peña and Santos, 2021; Wagner et al., 2019; Wise et al., 2022). In (Banta et al., 2002), sometimes the LLJ observations suggested the flow followed the terrain, whereas sometimes it remained at a constant height above mean sea level (AMSL). Other investigations also pointed out to the terrain-induced variability in wind speed in stable conditions (Mahrt et al., 2021; Radünz et al., 2020).

The terrain-induced variability in wind speed during SBLs and LLJs can lead to important spatial variations in power performance over complex terrain (Radünz et al., 2021, 2022). Other physical processes that can modulate farm performance in SBLs include wind farm wakes (Doosttalab et al., 2020; Gadde and Stevens, 2021) and blockage (Wu and Porté-Agel, 2017; Bleeg et al., 2018; Sebastiani et al., 2021; Schneemann et al., 2021; Sanchez Gomez et al., 2022). In Radünz et al. (2021), wind farms built in complex terrain were investigated because the back rows produced at times twice as much power as the front rows, despite the downwind wake effects undermining performance in the back rows. This performance pattern was associated with a strong downwind flow acceleration in stable conditions. The back rows were closer to the lee of the plateau, and thus had stronger winds available in comparison with the front rows. Later on, the occurrence of nocturnal jets and the depth of the stable layer were shown to amplify the horizontal variability in the winds and turbine performance (Radünz et al., 2022).

Specific attributes of the LLJ may make it particularly susceptible to terrain-induced accelerations, even in the presence of simple topographic features, potentially causing substantial changes in wind speed and wind farm performance. However, most existing research on LLJ interactions with wind farms has relied on idealized numerical simulations, leaving a critical gap in the understanding of real-world interactions involving LLJs, terrain, and operational wind farms. This gap underscores the need for field measurements and operational data to validate and improve models. The American WAKe ExperimeNt (AWAKEN), an international wind energy science project funded by the United States Department of Energy (DOE) and led by the National Renewable Energy Laboratory (NREL), aims to address this gap by advancing the physical understanding and modeling of atmosphere–wind farm interactions (Debnath et al., 2022; Moriarty et al., 2024). The study site, located in the U.S. Southern Great Plains (SGP) of northern Oklahoma, was chosen due to its dense concentration of wind farms, the availability of high-quality historical observations from the Atmospheric Radiation Measurement (ARM) Program's SGP facility (Krishnamurthy et al., 2021a), and the frequent occurrence of meteorological phenomena of interest for wind energy, such as southerly LLJs (Banta et al., 2002; Banta, 2008; Klein et al., 2015; Smith et al., 2019a; Krishnamurthy et al., 2021b). This field campaign provides an excellent setting for evaluating the two key goals of our investigation: (1) assessing whether simple terrain can induce significant spatial variability in wind speed, thereby influencing turbine wakes and performance, and (2) determining whether specific LLJ characteristics amplify terrain-induced spatial wind variability.





This paper investigates the interaction between a southerly nocturnal LLJ on 3 April 2023 and a seventeen-turbine subset of the King Plains wind farm, the most heavily instrumented site in the AWAKEN project. The terrain at the site is neither traditionally complex (e.g., mountainous or hilly) nor entirely flat, consisting of shallow river valleys with elevation changes of less than 50 m–referred to here as simple terrain. The remainder of the paper is organized as follows. Section 2 describes the orography and land use characteristics of the AWAKEN site, the observational dataset, criteria for case selection, and the simulation setup. The simulations use a multiscale WRF-LES-GAD (generalized actuator disk) approach, realistically driven by High-Resolution Rapid Refresh (HRRR) v4 analysis data. Section 3 integrates observations from ground-based scanning and profiling lidars, an atmospheric emitted radiance interferometer, and a sonic anemometer to validate the simulations and analyze the planetary boundary layer (PBL) winds, stability, and turbulence across the wind farm. Turbine performance is assessed using supervisory control and data acquisition (SCADA) system data. Section 4 provides a conceptual description of terrain-induced wind variability and compares its significance with wake and blockage effects. It also discusses the implications of terrain effects for multiscale modeling and wind farm control, as well as the potential long-term manifestation of terrain-induced variability associated with LLJs. Finally, Section 5 presents the main conclusions and outlines directions for future research.

## 2 Methods

### 2.1 The AWAKEN field campaign observations and case study selection

Five wind farms were selected for the AWAKEN field campaign, which are King Plains, Armadillo Flats, Breckenridge, Chisholm View, and Thunder Ranch (Fig. 1). This investigation focuses on the eastern portion of the King Plains wind farm because it is the most well instrumented location in AWAKEN. Furthermore, in the prevailing southerly winds (Debnath et al., 2022; Moriarty et al., 2024), it is not directly downwind of other wind farms. Therefore, the power performance patterns in eastern King Plains are solely caused by the terrain effects and turbine interactions that belong to that wind farm. We focus on a subset of 17 turbines organized into three rows.

The observations from the field campaign enable testing several scientific hypotheses, four of which are related to our work: the (i) wind farm wakes propagation (Bodini et al., 2024; Cheung et al., 2023; Krishnamurthy et al., 2024), (ii) blockage (Cheung et al., 2023; Sanchez Gomez et al., 2022), (iii) wake steering and (iv) individual turbine wake morphology. The AWAKEN site has relatively simple terrain, such that even the names of the wind farms bring this feature: King *Plains*, Armadillo *Flats*. Thus, understanding the terrain-induced spatial variability and influence of LLJs is relevant to the aforementioned scopes.

Observations from the A1 and A2 sites characterize the inflow because they contain no interference from nearby wind farms (Fig. 2) and are used for the case selection and model validation. At site A1, immediately upwind of the King Plains wind farm in the southerly wind direction, wind speed components were measured with a Halo scanning lidar (Letizia and Bodini, 2023). Reynolds stresses from this lidar were estimated based on the six-beam method of Sathe et al. (2015). Also at site A1, a Windcube v. 2 profiling lidar measured the wind speed components with a sampling rate between 0.5 to 1 s between 40 and 240 m above ground level (AGL) with a vertical resolution of 20 m (Wharton, 2023). A sonic anemometer at an upwind site


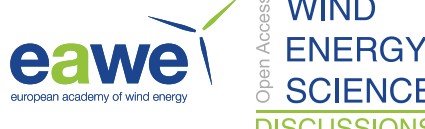

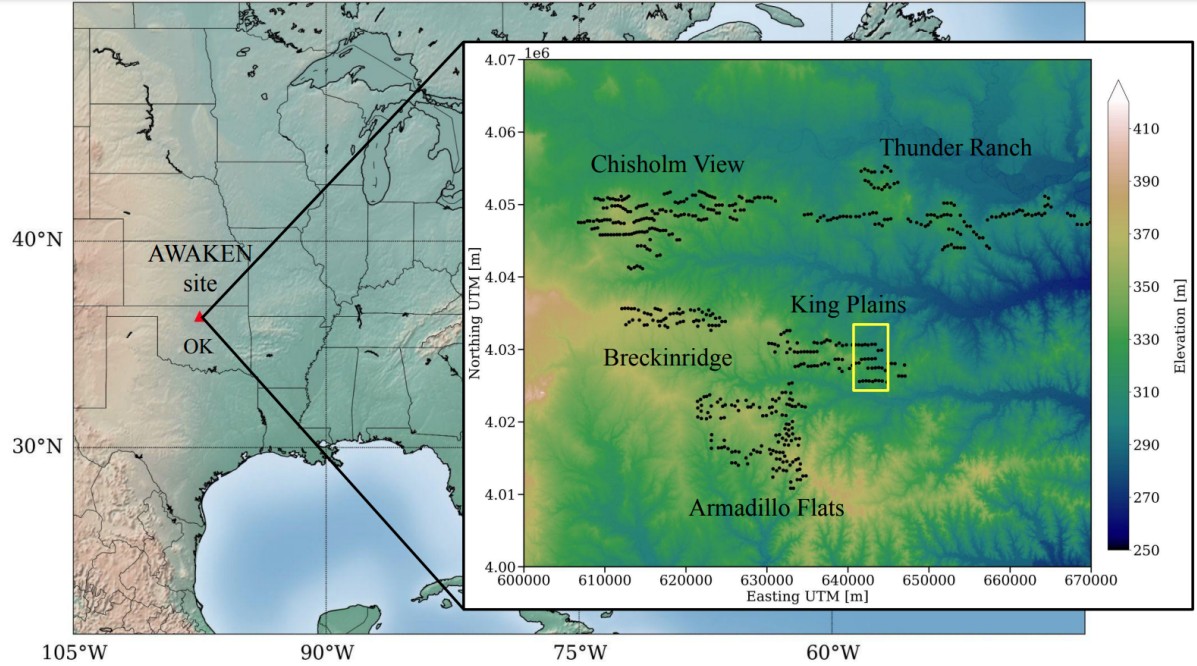

**Figure 1.** The background map of the United States of America (USA) shows the location of the AWAKEN site in Oklahoma (OK) near [36.5 °N, −97.5 °W]. In the foreground, the five wind farms that integrate the AWAKEN field campaign are shown overlaid on the terrain elevation map: King Plains, Armadillo Flats, Breckenridge, Chisholm View, and Thunder Ranch. The eastern portion of the King Plains wind farm is the focus of our investigation (yellow rectangle).

A2 measured wind speed components at a sampling rate of 20 Hz at 4 m AGL (Pekour, 2023). Although a profiling lidar was also located at A2, its poor data quality from 04:00 to 06:00 UTC during this case study was insufficient for analysis. Potential temperature profiles at site C1 (Atmospheric Radiation Measurement Southern Great Plains Central Facility) were derived from ground-based atmospheric emitted radiance interferometer (AERI) observations (Shippert and Zhang, 2016) using the TROPoe retrieval algorithm (Turner and Löhnert, 2014; Turner and Blumberg, 2019).

The case study consists of a southerly LLJ during a somewhat stationary window to mitigate the influence of larger-scale dynamic events in the analysis. Also, we selected nights with sufficiently strong winds to create turbulent motions resolvable with a $dx =$5 m grid (Skamarock, 2004), but not as strong as to cause turbines to operate near rated capacity, and thus impair our capacity to assess spatial variability in performance. Nights with weak winds create relatively small-scale turbulent motions that require a fine computational grid to be resolved. We also wanted stationary and moderate values for the sensible heat flux to avoid changes in the surface forcing. There were no requirements for the spatial variability in winds and performance. The case selection considered the following requirements. During the nighttime between 04:00 and 12:00 UTC, the time-averaged (i) wind speed at 100 m AGL should be between 5 and 14 m s$^{-1}$, (ii) wind direction at 100 m AGL between 160° and 200°,



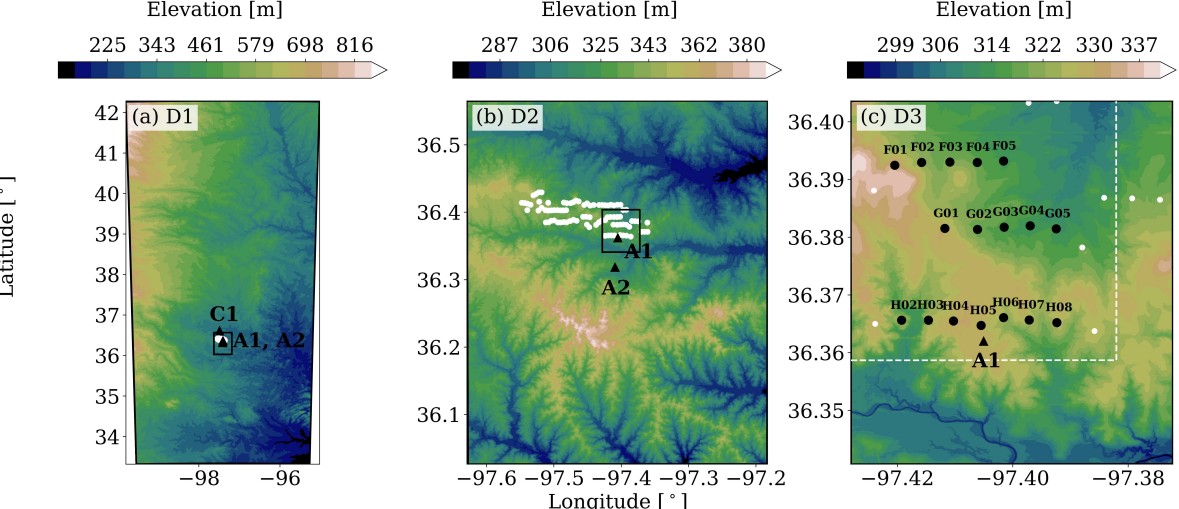

**Figure 2.** Horizontal extent and terrain elevation maps associated with the WRF simulations domains D1 (a), D2 (b) and D3 (c). The King Plains wind farm consists of 88 turbines (white dots), from which a subset of 17 wind turbines was represented in the simulation (black dots). Within domain D3, some turbines near boundaries were removed (white dots). Inflow fetches of 2 and 1 km in the southerly and easterly boundaries, respectively, were removed from the analyses (white dashed lines). The observation sites A1, A2 and C1 (black triangles) are also shown.

(iii) the standard deviation in wind direction at 100 m AGL below $50°$, (iv) the sensible heat flux lower than $-20$ W m$^{-2}$ and the (v) friction velocity lower than 0.5 m s$^{-1}$.

The dates from the year of 2023 that meet these criteria are 3 April, 18 April and 15 May 2023. Ultimately, 3 April 2023 was selected because of the predominant southerly wind direction ($180°$) and the moderate winds with large sensible heat flux, suggesting vigorous nighttime turbulence. The selected analysis window was from 04:50 to 05:25 UTC because of curtailment events at other times and occurrence of clouds in the simulations later on at 08:00 UTC.

## 2.2 Simulation setup

The multi-scale simulations were performed with the Advanced-Research Weather Research and Forecasting (WRF) model version 4.1.5 (Skamarock et al., 2019), which solves the compressible Euler equations for the three spatial dimensions and time. Three computational domains with a progressive increase in spatial resolutions and smaller areas were used to represent and bridge the large-scale, mesoscale, and microscale atmospheric processes (Fig. 2). The grid details specific to each domain are described in Table 1. For instance, the outermost domain (D1) had 201, 501 and 101 cells in the $x$ (west to east), $y$ (south to north) and $z$ (vertical) directions with a fixed horizontal resolution of $\Delta x = \Delta y = 2000$ m. In the vertical direction, the grid consisted of a finer and near-constant layer of $\Delta z_{sfc} = 30$ m in the first 1 km AGL, which was stretched out to a maximum of $\Delta z = 300$ m at the domain top. The large jump in spatial resolution between domains D1 ($\Delta x = 2000$ m) and D2 ($\Delta x = 100$ m)



**Table 1.** Computational grid and temporal information by domain.

| Parameter | D1 | D2 | D3 |
|---|---|---|---|
| $\Delta x, \Delta y$ [m] | 2000 | 100 | 5 |
| $\Delta z_{sfc}$ [m] | 30 | 20 | 4 |
| $n_x$ | 201 | 401 | 1001 |
| $n_y$ | 501 | 601 | 1401 |
| $n_z$ | 101 | 121 | 168 |
| Start time | 2 April 2023 12:00 UTC | 3 April 2023 03:00 UTC | 3 April 2023 04:30 UTC |
| End time | 3 April 2023 05:25 UTC | 3 April 2023 05:25 UTC | 3 April 2023 05:25 UTC |
| Time step [s] | 12 | 0.6 | 0.025 |

was intended to skip over the *terra incognita* (Wyngaard, 2004), where turbulence length scales are of the same order as the horizontal grid spacing ($100 < \Delta x < 1000$ m) (Muñoz-Esparza et al., 2017; Rai et al., 2019), as in the multiscale simulations

of Muñoz-Esparza et al. (2017). The innermost nest domain (D3) had a fixed horizontal resolution of $\Delta x = 5$ m. Each domain used a different vertical grid (vertical nesting (Daniels et al., 2016)), and the innermost domains were finer, such as done in

other multiscale simulations (Wise et al., 2022; Sanchez Gomez et al., 2022; Wagner et al., 2019).

      The static datasets employed for the terrain elevation and land use categories were the United States Geological Survey

(USGS) 1/3 arc-second ($\Delta x \approx 10$ m) dataset (U.S. Geological Survey., 2020) and the National Land Cover Dataset 2019 1 arc-second ($\Delta x \approx 30$ m) dataset (U.S. Geological Survey., 2019), respectively. The coarser Global Multi-resolution Terrain

Elevation Data 2010 (Danielson and Gesch, 2011) with a resolution of 30 arc-second ($\Delta x \approx 1000$ m) was used for domain D1.

      The terrain features can be described in terms of various scales. The domain D1 covers the foothills east of the Rocky

Mountains to the west and the sloping terrain that includes the SGP in the center and eastern regions (Fig. 2a). The domain D2 demonstrates that the terrain near the AWAKEN site is characterized by small river valleys and ridge lines with a domain-wise

maximum variation in elevation (amplitude) of 118 m (Fig. 2b). The innermost domain D3 displays the microscale terrain features in the King Plains wind farm area, with a domain-wise elevation amplitude of 49 m (Fig. 2c). The succession of the

small river valley depression near the southern boundary, the smooth ridge line and the beginning of the down-slope area that leads to the larger river valley further north appear at higher resolution. A subset of 17 turbines is represented in the finest

domain of the simulations. The turbines in the first row (H02–H08) are sited over slightly higher ground ($z \sim 330$ m AMSL), and those in the second (G01–G05) and third (F01–F05) rows are sited in the small down-slope area ($z \sim 320$ m AMSL),

which is shown in detail in the Appendix (Fig.A1c). Because of microscale terrain features, turbines H02, H03 and H06–H08 are located at local lower ground, and turbine F01 is at higher ground.

The initial and boundary conditions were provided by the High-Resolution Rapid Refresh (HRRR) model v4 (Dowell et al., 2022; National Oceanic and Atmospheric Administration (NOAA), 2024), with a horizontal spatial and temporal resolutions

of 3 km and 1 h, respectively. In this case study, simulations forced by the European Centre for Medium-Range Weather Forecasts Reanalysis v5 (ERA5) (Hersbach et al., 2020) and the North American Mesoscale Forecast System (NAM) (Rogers

et al., 2009) models data sets produced an exaggerated stratification and a weaker LLJ (Radünz et al., 2023), although other





studies of LLJ in this region (Smith et al., 2019a, b) have found success with NAM. The WRF simulation was from 2 April
2023 12:00 UTC to 3 April 2023 05:25 UTC, allowing 12 hours of spin-up time before the evening transition period at 00:00
UTC (Table 1). The domains were activated sequentially during the spin-up time. Considering an estimated 20-minute spin-up
time for the D3 domain, the analysis period was between 3 April 2023 04:50 and 05:25 UTC (35 min).

Several processes were included via parameterization schemes, such as for cloud microphysics, radiation, and the exchange
of momentum, heat, and moisture with the land surface. The main physics options adopted for each domain are summarized in
Table 2. All domains used the WRF Single-Moment 3-class simple ice scheme (Hong et al., 2004) for cloud microphysics, the
RRTM (Mlawer et al., 1997) scheme for short- and longwave radiation processes, and the Noah land surface model (Chen and
Dudhia, 2001). No cumulus parameterization option was included in any domain due to the fine resolution.

**Table 2.** Physics parameterization by domain. The model references are found in the text.

| Physics | D1 | D2 | D3 |
|---|---|---|---|
| Cumulus | – | – | – |
| Microphysics | WSM3 | WSM3 | WSM3 |
| Longwave radiation | RRTM | RRTM | RRTM |
| Shortwave radiation | RRTM | RRTM | RRTM |
| Land surface | Noah | Noah | Noah |
| Surface layer | MYJ | MYJ | MYJ |
| Planetary boundary layer | MYJ | – | – |
| LES SGS | – | 1.5TKE | NBA2 |
| CPM | – | – | on |
| GAD | – | – | on |

Turbulent processes in the planetary boundary layer (PBL) were accounted for differently in mesoscale and microscale
domains. In the outermost domain D1, turbulence was modeled using the Mellor–Yamada–Janjic (MYJ) PBL parameterization
scheme (Janjić, 1990; Janić, 2001), such as in other LLJ studies in the SGP (Storm et al., 2009; Storm and Basu, 2010;
Vanderwende et al., 2015). The Mellor–Yamada–Janjic (MYJ) surface-layer (SL) scheme was employed for all domains. In
the nested domains (D2 and D3), LES resolves turbulent motions larger than the grid size, and the influence of the interaction
between subgrid and resolved-grid motions on the resolved-grid flow field were modeled via subgrid-scale (SGS) models. As
in Zhou and Chow (2014) and Wise et al. (2024), different SGS models were used for the coarse and the fine LES domains.
The Deardorff 1.5 TKE SGS model (Deardorff, 1980) was used for domain D2. The nonlinear backscatter and anisotropy
(NBA) SGS model (Kosović, 1997), which accounts for backscatter and can improve turbulence dynamics, was used for the
innermost domain D3. SGS models that include backscatter are useful in multiscale simulations of stratified flows (Zhou and
Chow, 2014; Sanchez Gomez et al., 2022; Wise et al., 2024).

To accelerate the spin-up of turbulence within the innermost domain, we applied the cell perturbation method (CPM)
(Muñoz-Esparza et al., 2014, 2015; Muñoz-Esparza and Kosovic, 2018) in the southern and eastern boundaries. The CPM
applies random perturbations to the potential temperature field to trigger buoyancy fluctuations and shorten the fetch required



for turbulence spin-up. The current implementation in WRF dynamically uses information from the diagnosed PBL height
from the mesoscale domain to confine perturbations vertically. The potential temperature perturbation amplitude was calcu-
lated based on Muñoz-Esparza and Kosovic (2018) for a turbulent Eckert number ($E_c$) of 0.2. The CPM was deactivated for
domain D2 because it was too coarse to resolve turbulence in stable conditions, the same rationale used in (Wagner et al.,
2019). The turbulence was sufficiently spun-up between 1.5 and 2 km from the southerly boundary of domain D3 (Fig. B1 in
the Appendix). Thus, the analysis considers as a fully spun-up region a domain subset of 4 by 5 km, where 2 km and 1 km are
removed from the southerly and easterly boundaries of domain D3, respectively (Fig. 2c).

Wind turbines were represented in the simulations as generalized actuator disks (GAD) (Mirocha et al., 2014; Aitken et al.,
2014; Arthur et al., 2020). The GAD computes the axial and tangential forces imparted by the turbine to the flow based on
the aerodynamic properties of the blade and turbine control strategy. To circumvent the need for proprietary data, the GE 2.8-
127 turbine model installed in King Plains was emulated using an OpenFAST model. Publicly available turbines were used
as a template, and characteristics were tuned to match the GE 2.8-127 power and thrust curves (Quon, 2022). In addition to
the simulation with the wind turbines, a second simulation with the same configuration but without the wind turbines was
performed. This procedure enabled isolating the turbine wake effects.

## 3    Results

### 3.1    Wind, stability and turbulence in the planetary boundary layer

To assess the simulation skill in representing relevant rotor layer and surface quantities, a time series of selected variables
is compared against observations from a profiling lidar at site A1 and a nearby sonic anemometer at site A2 at 4 m AGL
(Fig. 3). A 10 minute rolling window is used to compute the instantaneous turbulence-related variables sensible heat flux ($H_s$)
and vertical velocity variance ($\overline{w'w'}$). The gray shaded area in Fig. 3 marks the 20 minute spin-up time, and the analysis is
conducted during the period between 04:50 and 05:25 UTC. The time averaged summary of the inflow conditions are shown in
Table 3. The subscripts in the variables denote the height AGL. The variables are the wind speed ($WS_{90}$ and direction ($WD_{90}$
at 90 m AGL, wind shear exponent between 40 and 160 m AGL ($\alpha_{40-160}$), wind veer between 40 and 160 m AGL expressed in
degree every 100 m ($\beta_{40-160}$), sensible heat flux at the surface ($H_s$) and vertical wind velocity variance at 90 m AGL ($\overline{w'w'}_{90}$).

**Table 3.** Time averaged values associated with the wind inflow of the 3 April 2023 case study between 04:50 and 05:25 UTC.

| Source | $WS_{90}$ [m s$^{-1}$] | $WD_{90}$ [°] | $\alpha_{40-160}$ [−] | $\beta_{40-160}$ [° 100 m$^{-1}$] | $H_s$ [W m$^{-2}$] | $\overline{w'w'}_{90}$ [m$^2$ s$^{-2}$] |
|---|---|---|---|---|---|---|
| WRF-A1 (D3) | 8.22 | 167 | 0.094 | 4.4 | −43 | 0.23 |
| OBS-PL-A1 | 8.09 | 167 | 0.170 | 7.5 | – | 0.21 |
| OBS-SNC-A2 | – | – | – | – | −57 | – |

The wind speed ($\approx 8.22$ m s$^{-1}$) and direction ($\approx 167$ °) remain in the range between 5–10 m s$^{-1}$ and 160–175° (southerly
winds), respectively, matching well the observations (Fig. 3a and b). Again, the reader may refer to Table 3 to verify the
average values. The wind shear exponent (Fig. 3c, calculated between 40 m and 160 m) is underestimated in the simulations



**Figure 3.** Time series of wind speed ($WS_{90}$, a) and direction ($WD_{90}$, b) at 90 m AGL, wind shear exponent between 40 and 160 m AGL ($\alpha_{40-160}$, c), sensible heat flux at the surface ($H_s$, d) and vertical wind velocity variance at 90 m AGL ($\overline{w'w'}_{90}$, e). The subscripts in the variables denote the height AGL. A 10-minute rolling window is used to compute $H_s$ and $\overline{w'w'}$. Observations from the profiling lidar at site A1 (OBS-A1-PL) and the sonic anemometer at site A2 (OBS-A2-SNC) are represented as continuous and dashed black lines, respectively. Simulation results for domain D3 are represented as continuous blue lines. The gray shaded area marks the 20 minute spin-up time.





($\sim 0.094$) relative to the observations ($\sim 0.170$). Observed wind speed (shear) decreases (increased) between 05:00 and 05:20

UTC, which does not occur in the simulations. Noticeably, the temporal fluctuations in the wind speed and direction signals are similar in amplitude and frequency to the observed ones. As in Muñoz-Esparza et al. (2017), this behavior indicates that

the turbulent motions were consistently resolved in the simulations.

Simulated turbulence metrics are within the range of values observed. The turbulence-related metrics, sensible heat flux at

230 the surface (Fig. 3d) and vertical velocity variance at 90 m AGL (Fig. 3e), approach the observed values. The simulated heat flux time series is, on average, ($-43$ W m$^{-2}$), weaker than observed values ($-57$ W m$^{-2}$). The simulated vertical velocity

variance (0.23 m$^2$ s$^{-2}$) is, on average, close to the observed values (0.21 m$^2$ s$^{-2}$).

The vertical structure of the boundary layer associated with the LLJ computed in the simulations matches well the observed

time-averaged wind speed, direction and potential temperature profiles, with a few differences (Fig. 4a–c). The observed LLJ speed profile has a maximum of about 24 m s$^{-1}$ at roughly 500 m AGL with an almost linear increase with height from the

surface to the nose (Fig. 4a), such as described in Banta (2008). The simulation results displays a somewhat flatter nose, below which the wind shear is stronger than in the observations ($300 < z < 400$ m AGL). Between 26.5 and 300 m AGL, the wind

shear is slightly weaker than in the observations. The potential temperature profile is stably stratified as in the observations, but the stratification is weaker between 26.5 and 300 m AGL in the simulations (Fig. 4c). Conversely, the stratification in

the simulation is higher than in the observations between 300 and 400 m AGL. However, the interpretation of the observed potential temperature profile should be used with caution: AERI retrievals are affected by poorer vertical resolution away from

the ground compared to the LES (Turner and Löhnert, 2014), which could explain some differences with the simulations. Also, the potential temperature profile at the C1 site is merely a proxy for the stratification at the A1 site, since there are

no measurements of this type there. The magnitude and vertical variation with height of the wind direction (veer) is well represented. The wind direction veers considerably with height, by about 4.4 ° every 100 m near the rotor (40 to 160 m AGL).

Scanning and profiling lidar observations are consistent, although the profiling lidar displays more variability, likely due to the smaller measurement volume and shorter temporal averaging period (Robey and Lundquist, 2022).

The turbulence-related variables calculated are consistent with observations (Fig. 4d,e) and the expected behavior for strong LLJs (Banta, 2008; Klein et al., 2015) as the strong wind shear results in mechanically-generated turbulence. The turbulence

intensity (TI) was computed as the ratio between the standard deviation and the mean of the 10-minute wind speed. The $\overline{w'w'}$ profile displays strong turbulence above rotor bottom tip ($z < 26.5$ m AGL) and below 300 m AGL, with a maximum near

the rotor top tip ($z < 153.5$ m AGL). This enhanced mixing weakens the stratification in the aforementioned layer. Below $z = 26.5$ m, the simulated $\overline{w'w'}$ is small and reaches zero at the surface. The associated peak in TI near the surface was caused

by the very small wind speed. Thus, near the surface, the stratification increased. These features are consistent with a strong LLJ, whereby turbulence is continuously produced below the jet nose owing to mechanical shear (Banta, 2008; Klein et al.,

2015). Because shear is small near the jet nose, so is turbulence, as mean shear is the main driver of mechanical production of turbulent kinetic energy.



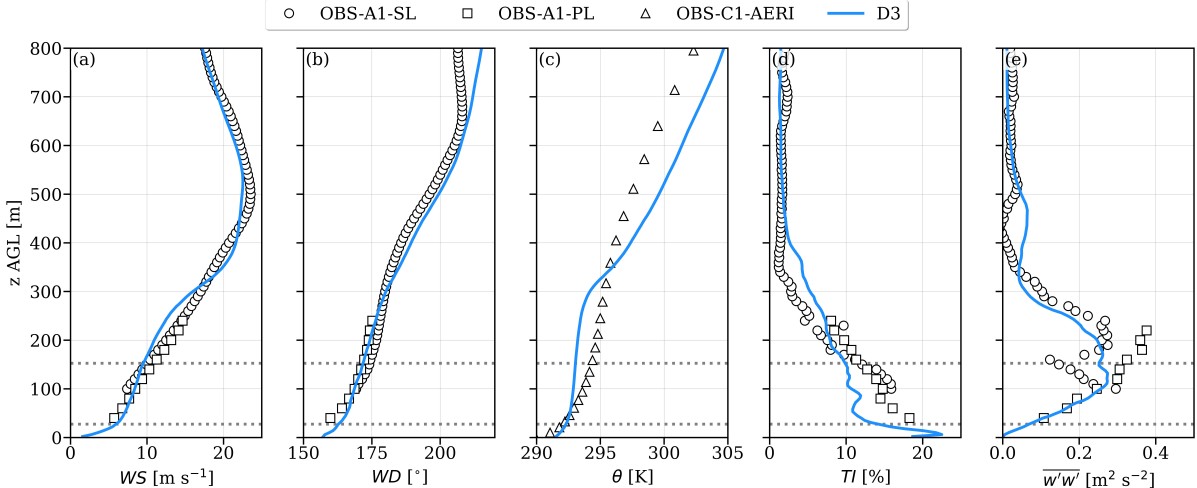

**Figure 4.** Vertical profiles of wind speed (a), direction (b), potential temperature (c), TI (d) and $\overline{w'w'}$ (e) for a 30 minute window between 04:55 and 05:25 UTC. Observations from the scanning lidar at site A1 (OBS-A1-SL), the profiling lidar at site A1 (OBS-A1-PL), and the AERI at site C1 (OBS-C1-AERI) are represented as markers. Results from domain D3 are represented as blue continuous lines.

Both the mean flow and the turbulence-related variables reasonably approximate the observations in the rotor layer and above. Therefore, the simulated flow field elsewhere is likely to approximate the behavior of the real flow field. Hence, we now examine the wind farm performance patterns and their relation to the flow field.

### 3.2 Wind farm performance

This section evaluates the power performance variability within the wind farm. The turbine power acquired with the supervisory control and data acquisition (SCADA) system is normalized as requested by the wind farm owner in the Non-Disclosure Agreement (NDA). The normalized mean power is computed as the ratio between the mean power of each row and the turbine rated power ($= P_{row}/P_{rated}$). The normalized mean power is separated into first (turbines H02–H08), second (turbines G01–G05) and third (turbines F01–F05) rows. For the southerly wind direction, the first row is upwind of the second and third rows.

An unexpected performance pattern occurs in the normalized mean power time series because the second and third rows consistently outperform the front row (Fig. 5) throughout this time period, even though the second and third rows should be impacted by wakes from the first row. This overperformance for the downwind rows occurs in both the observations and simulations. Further, the normalized mean power of the third row (SCADA = 0.937, WRF = 0.939) is slightly larger than that of the second row (SCADA = 0.776, WRF = 0.894). Both the mean power of the third and second rows are larger than that of the first row (SCADA = 0.621, WRF = 0.629). In percent, the first row is outperformed by the third (SCADA = 51 %, WRF = 51 %) and second (SCADA = 25 %, WRF = 44 %) rows. Although the simulated power of the second row is higher than that derived from the SCADA, the agreement for the first and third rows is better.



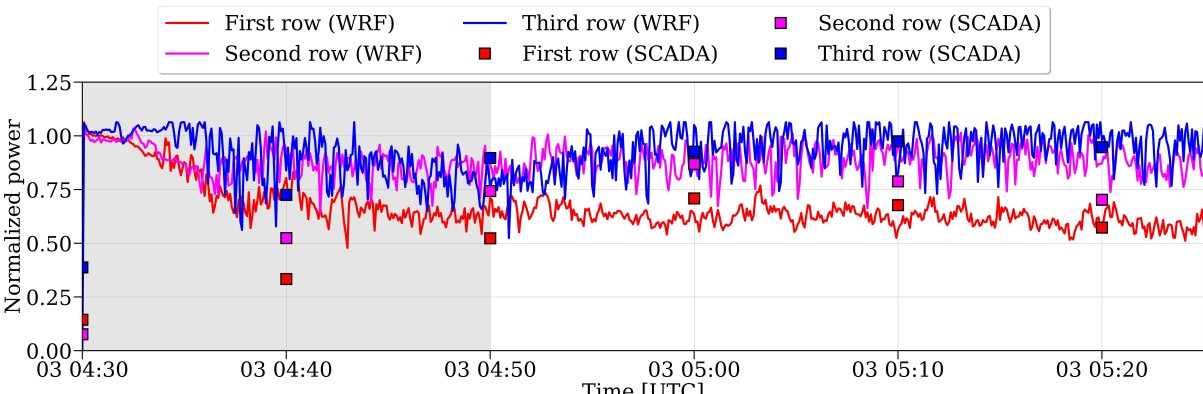

**Figure 5.** Time series of normalized mean power based on simulations (continuous lines) and SCADA data (markers). The gray shaded area marks the 20 minute spin-up time.

This performance differential was unexpected because the first row is free of wakes for the southerly wind direction and the second and third rows are likely to experience wake effects. Additionally, because of the gentle or simple terrain, the spatial

variability of winds was expected to be small. Given these circumstances, the first row was expected to outperform the back rows. Thus, some physical processes must have induced large spatial variability in the performance of the wind turbines over

relatively short distances (∼ 1–5 km), unrelated to wakes, and over simple terrain.

### 3.3    Instantaneous wind speed and wakes

This section assesses snapshots of the flow field to describe spatial variations in the mean flow and turbulence structures. Figure 6 shows the instantaneous wind speed maps at a fixed height of 90 m AGL at 05:00, 05:10 and 05:20 UTC for the

simulation with the turbines. The flow field over these 20 minutes contains similarities (Fig. 6a–c), such as the higher wind speed over the second and third rows and lower wind speed over the first row. Nonetheless, temporal variability in the flow field

also occurs. At 05:00 UTC, the front row has weaker winds in the westernmost turbines, stronger winds in the center, followed by another streak of weaker winds in the easternmost turbines. At subsequent times, the spatial variability consists of weaker

winds in the westernmost turbines and stronger winds in the easternmost turbines. However, despite the temporal variability in the flow field, the stronger winds over the second and third rows are sustained over time.

Considered in a vertical slice, the flow has three regions with distinct characteristics. Figure 7 shows the instantaneous wind speed and potential temperature at 04:54 UTC (11:54 LT) in a north-south vertical cross-section that roughly follows

the streamwise direction, for the simulation with and without turbines. First, a LLJ core region is near 800 m above mean sea level (AMSL) with very strong winds and weaker turbulence (Fig. 7a–b). At the bottom of the profile, a region with high shear

and vigorous turbulence exists between the ground level up to about 600 m AMSL. Finally, a layer with coherent turbulence structures induced by wind shear at the interface between both, which resemble Kelvin-Helmholtz Instabilities (KHIs), as in

Blumen et al. (2001); Zhou and Chow (2014) among others. The onset of KHI occurs after the second row, but has a more



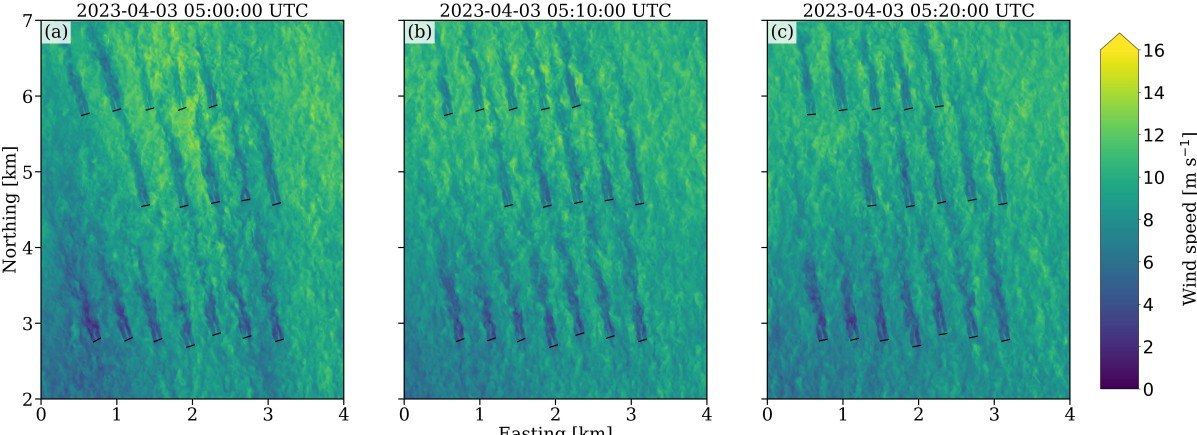

**Figure 6.** Instantaneous wind speed at 90 m AGL for domain D3 with the turbines at 05:00 (a), 05:10 (b) and 05:20 UTC (c) reveal that temporal fluctuations in the flow field co-exist with a sustained pattern of weaker (stronger) winds upwind (downwind). A video with the complete time window is provided as supplementary material (Radünz, 2024b).

salient structure near the third row (Fig. 7c–d). The vertical entrainment is most visible near the third row and below the LLJ
nose by assessing the potential temperature (Fig. 7c,d). The KHIs are not caused by the turbines because they also occur in
the simulation without turbines (Fig. 7b,d). However, qualitatively, the amplitude of the KHI appears to be amplified by the
presence of the turbines (Fig. 7c).

The vertical slices of wind speed and potential temperature reveal that the LLJ core region is displaced vertically (Fig. 7a–
d). Near the third row, the distance between the LLJ core region and the turbine rotor is smaller than near the first row. This
increase in wind speed associated with the downward displacement of the LLJ agrees with the spatial variability in wind speed
observed in the maps at a fixed height AGL (Fig. 6a–c).

## 3.4 Spatial variability in the mean wind speed and wakes

This section evaluates how the variability in the wind speed and wakes are sustained over time. Thus, the focus is the behavior
of the mean flow. First, we examine the horizontal variability in mean wind speed across different domains for the simulation
without turbines. The wider-scale domain D1 shows considerable spatial variability in wind speeds at 90 m AGL (Fig. 8a).
Domain D2 exhibits more detailed flow patterns (Fig. 8b), which includes an area with strong acceleration to the south of
domain D3 (Northing at about −10 km), followed by deceleration over the southern edge of the domain D3 (Northing at
about 0 km) and acceleration over the northern edge of the domain D3 (Northing at about 7 km). This pattern of acceleration-
deceleration-acceleration manifests across the whole of domains D1 and D2 due to terrain effects. At the highest resolution in
domain D3 and without turbines (Fig. 8c), the streamwise spatial variability in wind speed is between 6 m s$^{-1}$ in the southwest
and 12 m s$^{-1}$ in the north areas. In these stable conditions, the maximum wind speed does not always occur above the highest
elevation, but at a location slightly downwind of that elevation peak. Here, the strongest winds of about 12 m s$^{-1}$ occurred

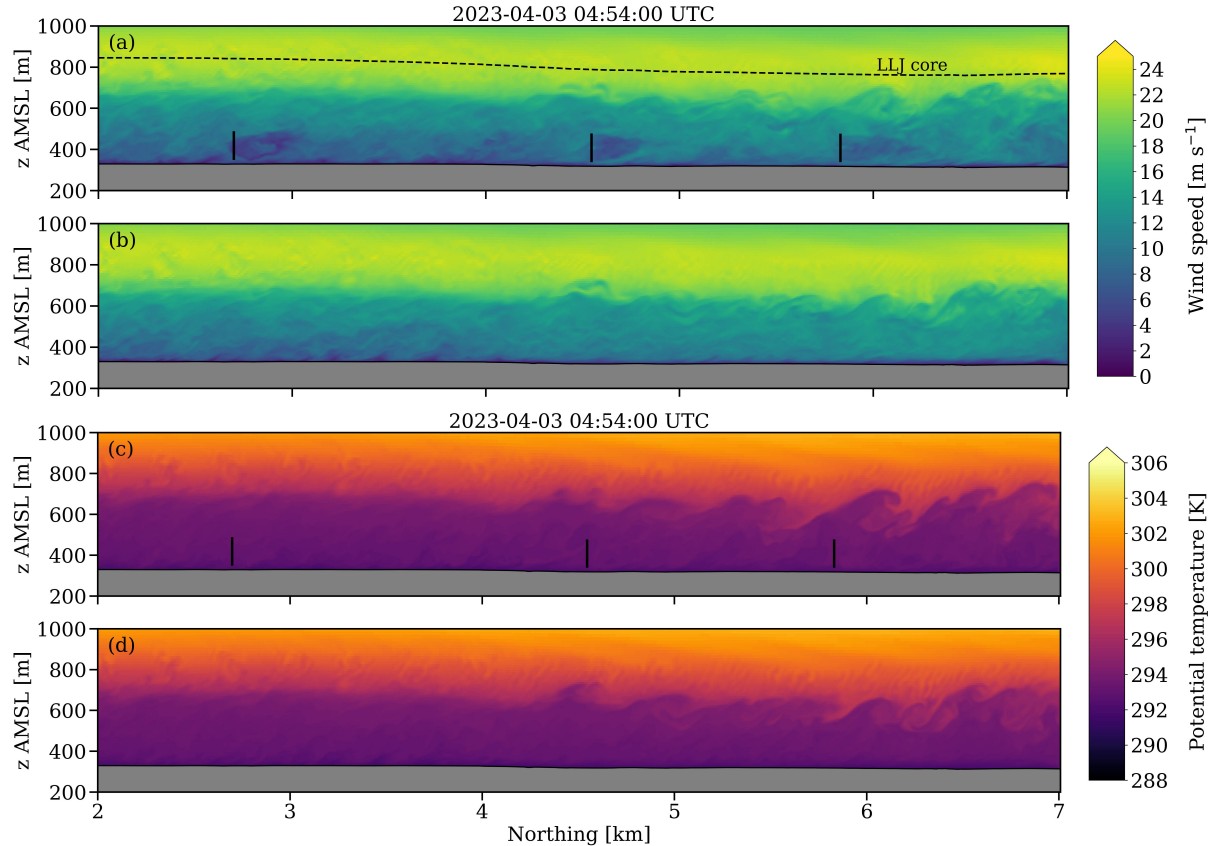

**Figure 7.** Vertical cross-sections in the south-north direction of wind speed (a, b) and potential temperature (c, d) for the simulation with (a, c) and without turbines (b, d). Videos with the complete time series of wind speed (Radünz, 2024d) and potential temperature (Radünz, 2024c) are provided as supplementary material.

over the eastern turbines of the third row (F03–F05), which are located at lower ground ($z \sim 320$ m AMSL) relative to the H05 turbine in the front row ($z \sim 330$ m AMSL), as shown in Fig. A1c. Notice that the turbine F01 from the third row is located at the highest elevation within domain D3 (Fig. 2c), but the winds over the third row are stronger near the turbines sited over lower ground (F03–F05). Thus, the relatively poor performance of the first row relative to the second and third rows is not driven by the upwind blockage effect (although this is certainly present), but by the terrain-induced spatial variability in wind speed.

Having demonstrated the influence of the terrain-induced spatial variability of winds on farm performance, we assess the variability in the wake effects. To delimit the extent of the wake, the time-averaged hub-height wind speed at a fixed height of 90 m AGL from the simulation without turbines (Fig. 9b) was subtracted from that of the simulation with turbines (Fig. 9a) and the wind speed deficits ($WS_{wt} - WS_{nowt}$) with magnitudes smaller than 1 m s$^{-1}$ were filtered out (Fig. 9c). The 1 m s$^{-1}$ threshold is adopted because a lower threshold (0.5 m s$^{-1}$) causes the wakes to merge, which complicates their individual





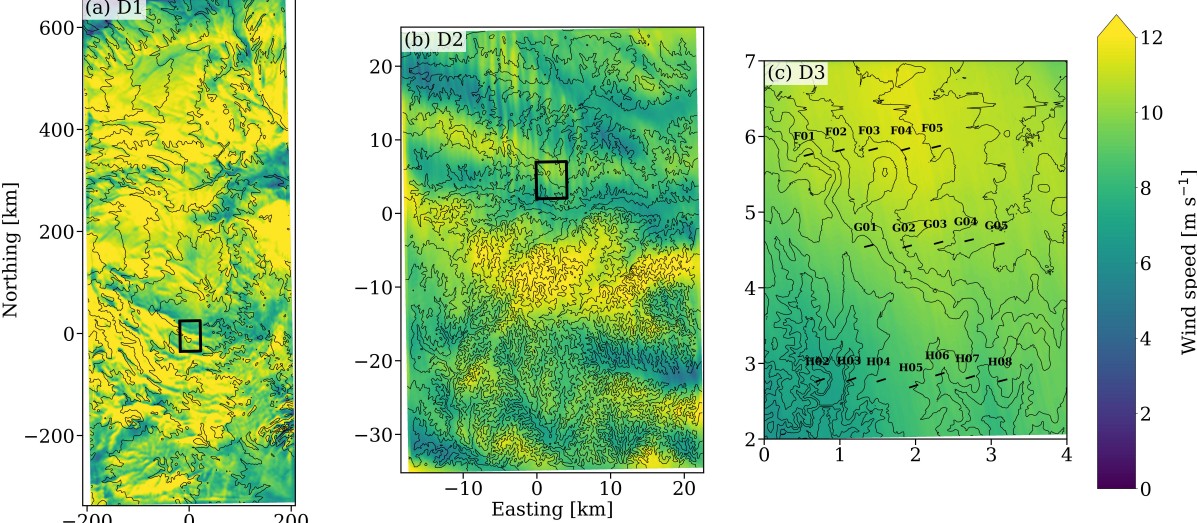

**Figure 8.** Time-average between 04:50 and 05:25 UTC of wind speed at 90 m AGL for domains D1 (left), D2 (center) and D3 (right) without turbines. The nested domain (black rectangle) and the vertical north-south cross-section (black line) are also illustrated.

assessment. The wakes from the first and second rows reach the downwind latitudes of the second and third rows, respectively.
The wake centerlines from turbines H06–H08 are not fully aligned with the rotor centerline of the downstream turbines. Thus, the wakes only partially reach the rotors with at least a 1 m s$^{-1}$ deficit. Conversely, the wakes from the second row flow in between the turbine rotors of the third row. Interestingly, the wakes of the H02 and H03 turbines in the front row are much shorter than the others (H04–H08) in the same row. This variability can be explained by the slower inflow wind speeds for H02 and H03, which at these wind speeds, considering the thrust coefficient variability of this turbine, produce a smaller thrust force and thus a weaker wake (Fig. C1a). Even if the thrust is lower in an absolute sense, H02–H03 are expected to have a higher coefficient of thrust ($C_t$) compared to the other generators in the domain and based on the $C_t$ vs $WS$ curve (Fig. C1b). Therefore, another possible explanation for the shorter wake of H02–H03 is the relatively stronger wake-added turbulence caused by the higher $C_t$ that can enhance wake recovery (Letizia and Iungo, 2022).

### 3.5 Vertical displacement and streamwise acceleration of the LLJ

This section analyzes how the interaction between the LLJ and the surrounding terrain leads to the observed horizontal gradients in wind speed. The multiscale influence of the terrain on the flow field is best pictured in north-south transects of wind speed (Fig. 10a–c) and vertical wind velocity (Fig. 10d–f) for domains D1, D2 and D3. The potential temperature isotherms aloft approximate the behavior of the mean flow streamlines and reveal undulations of various scales in the flow field, most clearly in domains D1 and D2. These undulations occur because, during the upslope flow, a low-level deceleration induces an upward component to the wind speed (positive vertical wind velocity) which displaces the LLJ core upwards. Conversely, during the downslope flow, a low-level acceleration occurs in phase with negative (subsiding) vertical wind velocity, and the LLJ



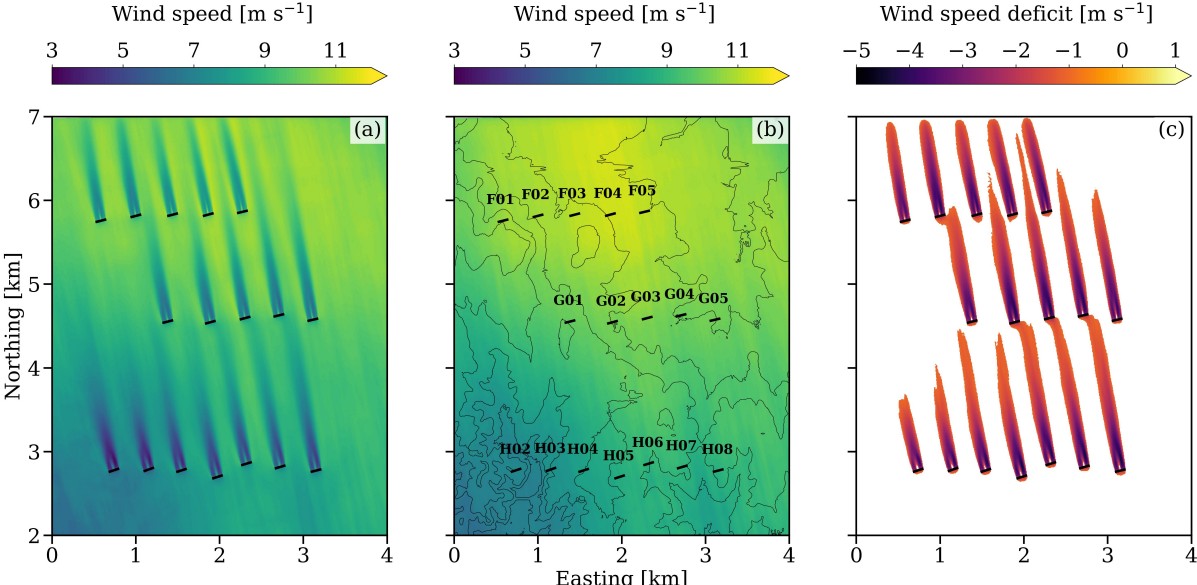

**Figure 9.** Time-average between 04:50 and 05:25 UTC of wind speed at hub height (90 m AGL) for the simulation with (a) and without (b) the turbines. The normalized wind speed deficit obtained by subtracting and normalizing the latter two indicates the extent of the wake region, considering a threshold of 1 m s$^{-1}$ (c).

core is displaced downwards. The buoyancy restoring forces associated with the initial, terrain-triggered up- and downward
motions create the undulations. The local maximum in wind speed occurs not at the local topographic peaks, but is shifted
downwind, typical of stable boundary layers (Baines, 1995). These peaks in acceleration are visible in the D1 domain near
$-300$ (P1), $-200$ (P2), $-60$ (P3) and $-7$ km (P4) (Fig. 10a), the latter being closest to King Plains. In domain D2 (Fig. 10b),
both the upwind acceleration near $-7$ km (P4) and a downwind acceleration near 6 km (P5) are identified, the latter being the
responsible for most of the wind speed spatial variability within the farm. Notice that the maximum wind speeds occur near
the transition between subsiding and ascending flow (see points P4 and P5 in Fig. 10b and e) because the wind is accelerating
during the downslope phase. Even though the flow field undulation appears small in the microscale domain D3 (Fig. 10c), it
brings the LLJ core down, closer to turbines in the second and third rows, enough to generate a measurable effect on power.
To better delineate the terrain-induced accelerations, the spatial variability in the wind speed is expressed as the difference
from a reference value, which is adopted as being the profile at the first row, $fr$. Thus, the wind speed difference ($WS - WS_{fr}$)
is essentially zero at the first row (Fig. 11a). A closer examination of the finest domain (Fig. 11a) reveals the streamwise wind
speed difference that can reach 3 m s$^{-1}$ over a distance of about 3 km downwind, forming a red layer below 300 m AGL.
Above the red layer near 500 m AGL, there is a white layer of either positive or negative but small wind speed differences.
Finally, near 600 m AGL there is a blue layer where the wind speed difference changes sign. An examination of the wind speed
differences at fixed heights AGL (Fig. 11b) reveals positive wind speed differences in the range between 2 and 3 m s$^{-1}$ over
the third row of turbines for heights below 300 m AGL.





**Figure 10.** Vertical cross-sections in the south-north direction of wind speed (a–c) and vertical wind velocity (d–f) with potential temperature isocontours for domains D1 (a, d), D2 (b, e) and D3 (c, f). The nested domain (thick black vertical lines) and projected positions of the H05, G02 and F04 turbine rotors (thin black lines) are also shown. The points P1 to P5 mark the local maximums in wind speed induced by the terrain.



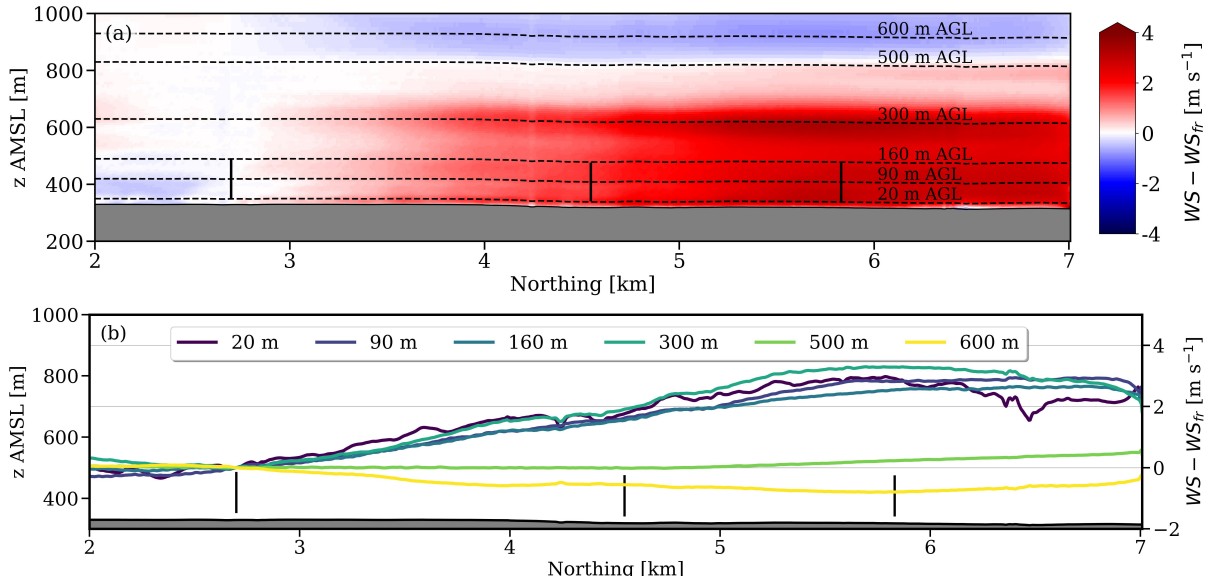

**Figure 11.** Vertical cross-section of wind speed difference relative to the front row profile ($WS - WS_{fr}$) for domain D3 in the simulation without turbines. The LLJ nose height is also displayed (a). Wind speed difference along terrain-following lines at fixed heights AGL (b).

Thus, three vertical layers with distinct wind speed difference patterns are identified above the second and third rows (Fig. 11a). First, the red layer with strong streamwise acceleration. Second, the white layer with near-zero streamwise ac-
celeration. Third, the blue layer above the LLJ nose with small negative wind speed difference. A plausible hypothesis for this pattern is the combination of vertical displacement and different sign and magnitude of the wind shear. For instance, the
downward displacement of the LLJ at a height AGL where the shear is large and positive (such as below the LLJ nose) will lead to streamwise acceleration at that height. However, the downward displacement at a height where the shear is large but
negative (such as above the LLJ nose) will lead to streamwise deceleration at that height. Likewise, if the shear is very small (such as near the LLJ nose) the vertical displacement will produce small changes in wind speed. We will refer to this as the
"rigid LLJ displacement" hypothesis. It assumes that the streamlines of the mean flow experience no change in wind speed, but only undulate because of the vertical displacement. Simultaneously, the actual flow acceleration along the streamlines is also
quantified. This approach enables assessing the individual contributions of both effects to the overall flow field acceleration at several heights (Fig. 12).

The streamlines are computed based on the bidimensional mean wind field using as starting points the heights between 20 and 600 m AGL near the inlet boundary (Fig. 12a). All streamlines are displaced downward ($\Delta z$) downstream of the first row
(Fig. 12b). The maximum displacement occurs near the third row at a distance of 6 km and increases with height AGL. The displacement varies between about $-15$ m (20 m AGL) and $-70$ m (500 m AGL), and stabilizes between 500 and 600 m AGL.
The actual wind speed differences along the streamlines ($\Delta WS_{str}$) are close to 2 m s$^{-1}$ at the lower levels ($< 160$ m AGL)





near the third row (Fig. 12c). At the higher levels near and above the LLJ nose (500 and 600 m AGL), the speed-up along the
380 streamlines is weaker ($< 1$ m s$^{-1}$).

The influence of the vertical displacement of the streamlines on the wind speed difference ($\Delta WS_{vert}$) is now assessed
(Fig. 12d). The vertical displacement produces small speed-ups ($\sim 1$ m s$^{-1}$) below 160 m AGL near the third now; a larger
speed-up ($\sim 3$ m s$^{-1}$) at 300 m AGL; a small slowdown at 500 m AGL ($> -1$ m s$^{-1}$); and a large slowdown at 600 m AGL
($\sim -2$ m s$^{-1}$). These variations in wind speed difference across streamwise distance and height AGL are explained by the
vertical variations in LLJ shear and magnitude of the downward displacement. For instance, at 300 m AGL the simulated wind
shear has a local maximum (Fig. 4a) and the downward displacement is relatively large ($\Delta z \sim -50$ m), which creates a large
$\Delta WS_{vert}$. Conversely, at 600 m AGL the simulated wind shear is negative (Fig. 4a) and the vertical displacement is even
larger ($\Delta z \sim -70$ m), which produces a large slowdown near the third row. Notice that downward displacement near the LLJ
at 500 m AGL ($\Delta z \sim -70$ m) is similar to that of 600 m AGL, but the small wind shear (Fig. 4a) leads to a small wind speed
difference. Thus far, the rigid LLJ hypothesis is corroborated.

If the conceptual description makes sense, the sum of the effects from the (i) rigid LLJ displacement and (ii) the actual
acceleration along the streamlines should produce speed-ups similar to those shown in Fig. 11b. Interestingly, the maximum
speed-up along the streamlines ($\Delta WS_{str} \sim 2$ m s$^{-1}$) occurs for the 20, 90 and 160 m AGL streamlines (Fig. 12c). The
394 streamlines higher up at 500 and 600 m AGL attain speed-ups between 0 and 1 m s$^{-1}$. Thus, the terrain-induced effects are
more pronounced at the lowest levels. When the vertical displacement (Fig. 12b) and the acceleration along the streamlines
(Fig. 12c) are combined (Fig. 11e), they lead to a maximum speed-up of about 3 m s$^{-1}$ for the streamlines at 20, 90 and
160 m AGL. This combined value of about 3 m s$^{-1}$ is close to the maximum speed-up previously obtained at fixed heights
AGL (Fig. 11b). The sum of $\Delta WS_{str}$ and $\Delta WS_{vert}$ also produces values consistent with Fig. 11b for the streamlines at 300,
500 and 600 m AGL.

Thus, both the (i) vertical displacement of the LLJ and the (ii) acceleration along the streamlines are important to the spatial
variability in wind speed. The difference is that whereas (i) occurs at all heights and is dependent on wind shear, (ii) exerts most
402 of its influence close to the ground. At 20 m AGL, about 18 % of the total speed-up is caused by the vertical displacement,
the rest being associated with acceleration along the streamlines. This ratio increases higher up at 90 m AGL (23 %) and
404 160 m AGL (39 %). At 300 m AGL, the vertical displacement is responsible for most of the overall speed-up. Thus, the vertical
displacement mechanism becomes more important with height because the vertical displacement also increases (Fig. 12b) and
406 the streamlines accelerate less (Fig. 12c) with height.

## 4   Discussion

### 4.1   The role of LLJs in the terrain-induced spatial variability of nocturnal flows

In stable conditions, the low-level wind decelerates upstream of obstacles more than it would in neutral conditions because
of the downward buoyancy (Mahrt and Larsen, 1990; Baines, 1995; Hunt et al., 1988) forcing the flow to stay at the same
altitude rather than rising over the obstacles. Likewise, the flow accelerates more in the lee, and this combination enhances



**Figure 12.** Vertical cross-section of wind speed difference relative to the front row profile ($WS - WS_{fr}$) for domain D3 in the simulation without turbines, with streamlines of the mean flow located between 20 and 600 m AGL near the inlet (a). The remaining subplots show the vertical displacement along the streamlines ($\Delta z$, b), actual change in wind speed along the streamlines ($\Delta WS_{str}$, c), estimated change in wind speed caused by the vertical displacement of the LLJ ($\Delta WS_{vert}$, d) and the estimated total change in wind speed of (c) and (d) combined ($\Delta WS_{tot}$, e). The aforementioned variables are relative to the value at the front row.



the spatial variability in wind speed. The $Nh/U$ parameter is commonly used in the literature to describe stratified flows over topography (Baines, 1995; Sauer et al., 2016; Fernando et al., 2019). Here, $N = \sqrt{\frac{g}{T_v}\frac{\Delta\theta_v}{\Delta z}}$ is the Brunt-Vaisälä frequency, $g$

is the gravity acceleration, $T_v$ is the virtual temperature, $\Delta\theta_v$ and $\Delta z$ are the difference in potential temperature and height between two vertical levels, respectively. Then, $h$ is the characteristic height of the topographic feature and $U$ is the wind

speed. This behavior of stratified flows over terrain occurs, to a lesser or greater extent, in any SBL, with LLJs representing a particular case. This section explores how LLJs contribute to larger spatial variations compared to regular SBLs.

One key factor in this enhanced spatial variability is the high wind shear characteristic of LLJs. In Section 3.5, we identified a mechanism whereby wind shear combined with downward displacement of the mean flow streamlines causes wind turbines

downstream to experience stronger winds than those upstream (Fig. 12a). Combining the magnitude of the downward displacement with a reference LLJ wind speed profile at the first row, which we denoted as the "rigid LLJ hypothesis", we demonstrated

that part of the spatial variability in wind speed is caused by the vertical variations in wind shear and the vertical displacement of streamlines. For instance, below the LLJ nose where the wind shear is high and positive (Fig. 4a), the downward displace-

ment leads to a streamwise speed-up (the red layer in Fig. 12a, heights below 300 m AGL in Fig. 12d). Conversely, near the LLJ nose the shear is near zero, so that the downward displacement produces very small speed-ups (the white layer in Fig. 12a,

at 500 m AGL in Fig. 12d).

      Notice that the vertical displacement of the streamlines in the SBL is an important component for the spatial variability in

wind speed. In this regard, the depth of the SBL plays a key role. Idealized numerical simulations revealed that for SBLs with the same wind speed and stratification, the case with a deeper layer produced a stronger acceleration in the lee of the obstacle

(Durran, 1986). Thus, the $Nh/U$ parameter was the same, with only a difference in $H_{SBL}/h$, where $H_{SBL}$ is the depth of the SBL. One interpretation is that the buoyancy force acts over the stratified layer and it being too shallow restrains the magnitude

of the acceleration. This feature was discussed in experimental and theoretical works on stratified flow and obstacles (Long, 1955; Baines, 1995). More recently, a field measurement campaign and numerical simulations revealed the evolution of strong

downslope flow in the Alaiz mountain (Santos et al., 2020; Peña and Santos, 2021). After the evening transition, the recently formed SBL developed into a LLJ. However, the spatial variability in the wind over the mountain becomes large only later

on when the SBL becomes deeper. Then, the wind accelerates down the slope and produces lee waves. A similar process was also described in the Perdigão field campaign, where the maximum wind speed was not at the top of topographic features, but

rather further downstream (Fernando et al., 2019). Likewise, a wind farm with two rows of turbines located in complex terrain experienced a similar nocturnal pattern in the wind (Radünz et al., 2021, 2022). On average, very stable conditions dominated

in the early nighttime, whereas weakly stable conditions dominated in the late nighttime (Radünz et al., 2021). However, the back rows outperformed the front rows consistently only in the late nighttime. That apparent inconsistency was attributed to

the SBL being deeper later on, despite being less stratified (Radünz et al., 2022). One important implication of this fact is that the LLJs later in the nighttime are likely associated with deeper SBLs and more prone to higher spatial variability. Our analysis

refers to 5 hours after the evening transition, which allows the SBL to develop a reasonable depth.

      The relatively weak winds near the surface, caused by the high wind shear, also could make LLJs more prone to large spatial

variability. In Zhou and Chow (2014), multiscale LES were used to investigate cold air pooling and a turbulence intermittency





mechanism. The valley was oriented in the east-west direction (similar to ours), whereas the winds were southerly. They
discussed that considering the relatively simple terrain (small $h$) and the relatively high wind speed (high $U$), the low $Nh/U$
value suggests that the cold air would be swept away from the valley towards the north. However, the valley shelters the cold
air from the stronger wind aloft, so that the cold pool remains. This rationale could be adapted to interpret our results. The
strong LLJ (nose wind speed >20 m s$^{-1}$) and the simple terrain ($h \sim 10$–$50$ m) would suggest small terrain-induced effects
(low $Nh/U$). Nonetheless, because of the high shear the wind speed is considerably smaller in the rotor layer (from 6 to
10 m s$^{-1}$) and the $Nh/U$ parameter is larger. Because of the higher $Nh/U$, the weaker surface winds are more susceptible
to deceleration up the slope, and consequently upper layers of the LLJ are displaced upwards. Conversely, the low-level wind
accelerates down the slope and the LLJ moves downwards. This process creates the spatial variability in wind speed (Fig. 11).
The flow acceleration along the streamlines, shown to be more pronounced near the ground and weaker higher up, is related to
this feature (Fig. 12b).

Therefore, aspects related to wind shear and SBL depth associated with the LLJ reported here, and with LLJs in general,
make it susceptible to terrain-induced accelerations. In that situation, even simple terrain can induce streamwise changes in
wind speed capable of significantly impacting wind farm performance patterns.

## 4.2 Comparison with other physical mechanisms that modulate wind farm performance in stable conditions

This section discusses how the terrain-induced spatial variability in wind speed compares with other physical processes that
influence farm performance in stable boundary layers, such as wind farm blockage and wakes. The literature discussion below
applies a spatial variability metric that measures the degree of horizontal wind speed variations associated with each process,
computed as the amplitude of the wind speed deviation or deficit caused by the process divided by a reference wind speed
($\Delta WS/WS_{ref}$).

The terrain-induced variability in wind speed reported here is about 4 m s$^{-1}$ over a distance of 5 km, or 50 % for a reference
wind speed of 8 m s$^{-1}$ (Fig. 11). Consequently, the second and third downwind rows of turbines produced between 20 and
50 % more power than the first row (Fig. 5). Based on the $Nh/U$ parameter, given the same inflow wind speed ($U$) and
stratification ($N$), the more complex terrain (higher $h$) likely amplifies the magnitude of the flow field variability relative to
simpler terrain (Baines, 1995). For instance, when $Nh/U$ increases from 1.56 to 2.84, that produces low-level flow blockage
and a more acute acceleration near the lee (Fig. 5.3 in Baines (1995)). At a wind farm built over a plateau with an elevation
change of 100–150 m upwind and 160–300 m downwind, a wind speed difference of the order of 3–4 m s$^{-1}$ was identified
between two rows of turbines separated by a distance of about 1 km (Radünz et al., 2021, 2022). That is the same order of
variability reported here ($\sim 4$ m s$^{-1}$ over 5 km) but over a much shorter distance. Not surprisingly, at times the turbines in the
back row in that investigation produced twice as much power as those in the front row.

In Liu and Stevens (2021), a contrasting mechanism for a terrain-induced enhancement in power performance during a LLJ
emerged. In the idealized LES simulation, a single wind turbine was positioned downstream of a single hill. The performance
enhancement was attributed to the higher momentum entrainment from the LLJ that was caused by the hill wake. We argue
that despite this mechanism being possible in other situations, during stable conditions the attached flow tends to minimize





hill wakes. Most importantly, the mechanism described by the linear wave theory and towing tank experiments (shallow water
equations) is unrelated to turbulent transport, but to modifications experienced by the mean flow (Baines, 1995). The terrain-
induced spatial variability in wind for stable boundary layers and LLJs literature (Santos et al., 2020; Peña and Santos, 2021;
Radünz et al., 2021, 2022) and the results we report on belong to the latter type (Baines, 1995).

There is consensus that wind turbine (Schepers et al., 2012; Abkar and Porté-Agel, 2015) and farm wakes (Barthelmie and
Jensen, 2010; Hansen et al., 2012; Abkar et al., 2016; Lundquist et al., 2018; Krishnamurthy et al., 2024; Porté-Agel et al.,
2020) recover slower in stable conditions because of the weaker entrainment of momentum into the wake. Idealized LES studies
report wind speed deficits between 20 and 30 % at downwind distances between $8D$ and $12D$ for a single wind turbine (Abkar
and Porté-Agel, 2015) and up to 50 % for a wind farm (Abkar et al., 2016) in stable conditions. Our results reveal a wind speed
deficit of about 1 m s$^{-1}$ ($\sim$ 12.5 %) over the second and third rows of turbines (Fig. 9). In part, differences can arise from
different power curves from turbine models and operating conditions. However, high turbulence levels (Hansen et al., 2012)
and veer (Abkar et al., 2016) are known to enhance wake recovery. To some extent, the somewhat weaker wakes reported
here could be a byproduct of the relatively high TKE, since this LLJ is strong (Banta, 2008; Klein et al., 2015), and wind
veer. Therefore, the terrain-induced spatial variability in wind speed of about 4 m s$^{-1}$ ($\sim$ 50 %) overshadows the variability
associated with the wake deficits of our study ($\sim$ 12.5 %) and scales with the deficits associated with idealized simulations of
wind farms in stable conditions ($< 50$ %) (Abkar et al., 2016).

The terrain-induced spatial variability in wind influences the variability in the region of operation of individual turbine
power curves (Fig. C1a,b). That leads to a spatial variability in the wakes (Fig. 9c) and can have an important outcome for
wind farm control, which is considered to be most useful in the stably stratified conditions such as considered here (Fleming
et al., 2019). Porté-Agel et al. (2020) discussed two aspects that modulate wind farm wakes and performance. The first is the
diurnal cycle and its modulation of stability, turbulence, shear and veer, discussed in the previous paragraph. The second is
the presence of terrain induces a non-zero streamwise pressure gradient, vertical displacement of the wake center, and flow
separation. Based on our work, it can be added that the interplay between terrain and stability, particularly in the case of LLJs
(Section 4.1), is an important source of wind farm wake variability, even for simple terrain. Wind farm control, regarded as one
of the three grand challenges associated with wind energy science, is intimately linked with another grand challenge related to
the physics of multiscale atmosphere-wind farm interactions (Veers et al., 2019, 2022). At the core of wind farm control are
the wake effects and strategies to manipulate their intensity and propagation in the wind farm area (Meyers et al., 2022). Wind
farm control was shown effective in idealized numerical studies and tests in real wind farms (Fleming et al., 2019; Simley
et al., 2021). Although there were power performance benefits in neutral and convective conditions, the benefits were most
pronounced in stable conditions. It was pointed out that challenges in complex terrain (associated with the spatial variability of
winds) should be addressed in future studies (Meyers et al., 2022). The AWAKEN project site was selected owing to the lesser
terrain complexity and has ongoing wind farm control studies (Moriarty et al., 2024). The existence of spatial variability in
wake effects, as revealed by turbines H02 and H03 (Fig. 9c), highlights that accounting for terrain effects, however simple the
terrain may be, is important for wind farm control. However, the spatial variability in wind causes certain turbines to operate in
Region 2, whereas others operate close to or in Region 3, imparting differences in thrust force (Fig. C1a,b) and thus the wake





deficit. Some multiscale models, such as AMR-Wind, do not yet consider topographic effects, and how this influence will be considered for model intercomparison and testing the AWAKEN scientific hypotheses is an open problem. Thus, wind farm

control strategies will likely be influenced by this terrain induced variability.

Previously, we discussed that the low-level deceleration upwind and acceleration downwind of topographic obstacles dis-

520 places the LLJ vertically, and is a source of spatial variability in wind speed (Section 4.1). A similar process occurs when wind farm wakes decelerate the wind speed and the LLJ is displaced vertically (Wu and Porté-Agel, 2017; Larsén and Fischereit,

2021; Krishnamurthy et al., 2024; Quint et al., 2024) even without the presence of terrain. In a numerical investigation of a conventionally-neutral boundary layer, the exit region of the wind farm displayed flow acceleration compared with the en-

trance region (Wu and Porté-Agel, 2017). Immediately downwind, the flow experienced further acceleration owing to (i) the boundary layer adjustment and (ii) the downward displacement of the stratified free-atmosphere (their Figure 4). Even though

a conventionally-neutral boundary layer is different from a SBL, the concept associated with the streamwise acceleration in the exit region and downwind is similar, the release of turbulent stresses induced by the "roughness" of the turbines. There-

fore, accounting for and quantifying the terrain-induced and wind-farm-induced spatial variabilities is important for wind farm performance assessments.

The spatial variability associated with terrain, demonstrated here, is an order of magnitude larger than that due to upwind blockage. According to the literature, the blockage effect may produce a maximum slowdown in the wind speed of about

10 % and often below 5 % (Wu and Porté-Agel, 2017; Bleeg et al., 2018; Sebastiani et al., 2021; Schneemann et al., 2021; Sanchez Gomez et al., 2022). Using Reynolds-averaged Navier-Stokes simulations with turbines represented as actuator disks,

Bleeg et al. (2018) showed that the blockage effect produced an average slowdown of 3.4 % of the free wind at a distance of two rotor diameters upwind, and an average slowdown of 1.9 % at seven to ten rotor diameters upwind. Another study used

multiscale simulations at the same King Plains wind farm during a LLJ episode (Sanchez Gomez et al., 2022). The slowdowns associated with the blockage effect varied between 8 % ($0.64\ \mathrm{m\ s^{-1}}$) and 1 % immediately upwind and $24D$ upwind of the first

row of turbines, respectively. At an offshore wind farm during stable conditions and with turbines operating in the high-thrust coefficient regime, the blockage was about 4 % upwind the wind farm (Schneemann et al., 2021). In Wu and Porté-Agel (2017),

the maximum blockage $2.5D$ upwind of the first row of turbine was 0.8 to 1.2 % (weak free-atmosphere stratification) and 10 to 11 % (strong free-atmosphere stratification). This means that the spatial variability in wind speed associated with blockage

($\sim$ 1–10 %) is likely one order of magnitude smaller than that of the terrain ($\sim$ 50 %).

Finally, whereas the farm performance is especially sensitive to wakes and blockage in high thrust coefficient conditions and

544 environmental conditions associated with strong stratification and low TI, the terrain-induced variability can affect performance from cut-in to near-rated wind speeds and during weaker stratification and high TI scenarios. These three effects will also

modulate performance differently across different wind speed, direction, stability and turbulence conditions. Also, the terrain complexity further enhances the terrain-induced variability.





## 5    Conclusions

Stratified flows over terrain have always attracted scientific interest from engineers and atmospheric scientists. The flow patterns that include lee waves and hydraulic jumps have been observed in controlled laboratory experiments as well as in field campaigns in the natural environment. Advances in wind energy now demand a deeper understanding of the behavior of stable boundary layers (SBLs) and nocturnal low-level jets (LLJs) as they interact with wind farms and the surrounding landscape. The scientific literature generally attributes challenges to *complex terrain*, but the results we report suggest that even sites that do not belong to that category, which we denote *simple terrain*, may experience important terrain-induced spatial variability in the wind resources.

Using both observations and multiscale numerical simulations of a strong LLJ at the AWAKEN campaign, we show that even simple terrain can induce important accelerations in the flow field. Furthermore, specific attributes of LLJs that may separate them from regular stable boundary layers when it comes to the terrain-induced spatial variability in wind speed. For a site in simple terrain in stable conditions, the expectation is that the downwind turbines will likely produce less energy than the upwind turbines because of the wake effects. However, here, results from a realistically-forced multiscale simulation with the WRF-LES-GAD approach were corroborated by the wind farm SCADA data to reveal that turbines in the front row were actually outperformed by those in the second (SCADA = 25 %, WRF = 44 %) and third (SCADA = 51 %, WRF = 51 %) rows. This performance variability occurred because of terrain-induced spatial variability in wind speeds, which produced stronger winds over the second and third rows. The relatively simple terrain, combined with certain attributes of the LLJ, induces a streamwise speed-up of up to 4 m s$^{-1}$ over a distance of 5 km. The mechanism underlying this acceleration is related to the vertical displacement of the LLJ combined with the high positive wind shear below its nose. Near the ground, the wind also decelerates or accelerates along the mean flow streamlines. Higher up, the acceleration along the streamlines is smaller, and the vertical displacement (undulation) of the streamlines (in response to low-level flow de/acceleration) causes the downstream wind speeds at the same height above ground level (AGL) to increase. Conversely, near the LLJ nose where the wind shear is near-zero, the vertical displacement produces small variations. Above the LLJ nose, where the wind shear is negative, the downward displacement produces small slowdowns.

The terrain-induced spatial variability in wind speed associated with the LLJ has implications to the wind farm performance and control literature. The aforementioned effect can not only be important at sites with apparently simple terrain, but also scale with or even overshadow the degree of variability associated with other mechanisms that modulate wind farm performance, such as turbine wakes and wind farm blockage. Wind farm control is regarded as one of the grand challenges in wind energy science. Thus, the horizontal gradients in wind speed induced by even simple terrain should be accurately represented in the numerical tools that implement farm control.

Future work should address how the long term terrain-induced spatial variability in wind speed is influenced by the diurnal cycle. Specifically, work could assess (i) differences between the unstable (typically daytime) and stable (typically nighttime) conditions, (ii) non-LLJ and LLJ stable cases and (iii) the role of the SBL depth and wind shear in the spatial variability in wind speed. This research direction is important because more statistical significance of unstable and stable boundary layer flows





will enable drawing contrasts on how they modulate spatial variability in wind speed. Furthermore, it can help to elucidate the key parameters that describe the spatial variability in wind speed. One of the main challenges is to distinguish the terrain-induced acceleration effects from other processes that also produce spatial variability in wind speed, such as wind farm wakes, blockage and dynamic events. If successful, however, this inquiry will lead to a deeper understanding of how physical processes in the atmospheric boundary layer influence wind farm flows and performance in the long term. These insights can guide the development of computational frameworks to improve the design and operation of modern wind farms.

*Code and data availability.* The AWAKEN observations are available online at the Data Archive Portal (DAP) at https://a2e.energy.gov/, except for the turbine SCADA data, which is proprietary. Links to the observations used in this paper can be accessed at Letizia and Bodini (2023); Pekour (2023); Shippert and Zhang (2016); Wharton (2023). The WRF-LES-GAD model framework version 4.3 is available at https://github.com/a2e-mmc/WRF/tree/mmc_update_v4.3 (which is not the same version we used). The WRF-LES-GAD model framework version 4.1.5 can be made available upon request. The terrain and land-use data can be downloaded at U.S. Geological Survey. (2020, 2019). The HRRR data can be downloaded at National Oceanic and Atmospheric Administration (NOAA) (2024). The WRF setup files for the two simulations can be accessed at Radünz (2024a).

*Video supplement.* Temporal evolution of streamwise transects of wind speed (Radünz, 2024d) and potential temperature (Radünz, 2024c). Temporal evolution of wind speed maps at 90 m AGL (Radünz, 2024b).

## Appendix A: Detailed terrain elevation profiles

Although the terrain might initially appear complex, the elevation variations occur over large horizontal distances. In domain D1 (Fig. A1a), the maximum variation in elevation is approximately 300 m ($\approx$ 500–200 m AMSL) over a horizontal distance of about 900 km. In domain D2 (Fig. A1b), two notable topographic features are evident. The first is a larger feature spanning $-30$ to 0 km, with a characteristic height ($h$) of about 60 m ($\approx$ 360–300 m AMSL). The second is a smaller feature spanning 0 to 10 km, with a characteristic height of about 30 m ($\approx$ 330–300 m AMSL).

Within the wind farm area (domain D3; Fig. A1c), elevation variations are even more subtle, with a maximum difference of about 10 m ($\approx$ 330–320 m AMSL) between the first and second/third rows of turbines. Despite this minimal elevation variation, the downstream rows experience stronger winds (Fig. 8c) and higher energy production (Fig. 5).

## Appendix B: Turbulence spin-up fetch

Turbulence is considered fully developed when the streamwise changes in the $\overline{w'w'}$ profile become relatively small. Near the inflow boundary (Northing = 0 km), turbulence is virtually absent, as indicated by $\overline{w'w'}$ values near zero at all heights (Fig. B1). At Northing 0.5 km, the $\overline{w'w'}$ profile exhibits overexcited turbulence compared to profiles at downstream locations



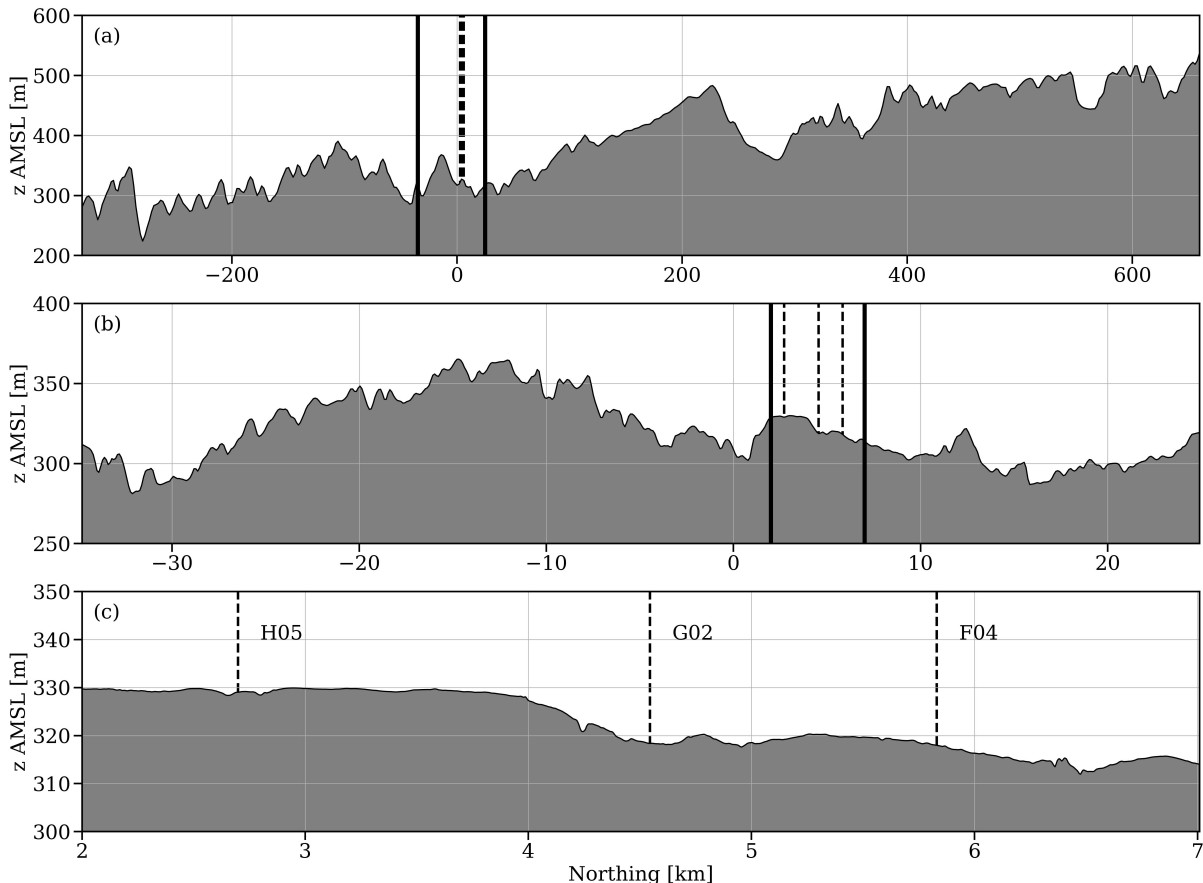

**Figure A1.** Terrain elevation profiles along a north-south transect for domains D1 (a), D2 (b), and D3 (c). The solid black vertical lines delimit the nested domains, while the dashed vertical lines mark the locations of turbines in the first (H05), second (G02), and third (F04) rows.

below 100 m AGL. Between Northing 1.5 km and 2 km, the $\overline{w'w'}$ profiles show better agreement, suggesting that turbulence has spun up. The $\overline{w'w'}$ profiles in this region also align closely with those observed at the A1 site, including observational data

(Fig. 4e). Consequently, we adopted a 2 km fetch as the spin-up region (Fig. 2c).

**Appendix C: Turbine power curves and mean wind speed**

The spatial variability in turbine performance is evident in Fig. C1a,b. The wind speed across turbines varies significantly, ranging from approximately 7 to 11 m s$^{-1}$, representing a difference of about 4 m s$^{-1}$. As a result, some turbines in the first

row (H02–H04) operate in Region 2 of the power curve, with generator power values between 1000 and 1500 kW (Fig. C1a). In contrast, turbines in the third row (F01–F05) operate near or at rated capacity (2820 kW).



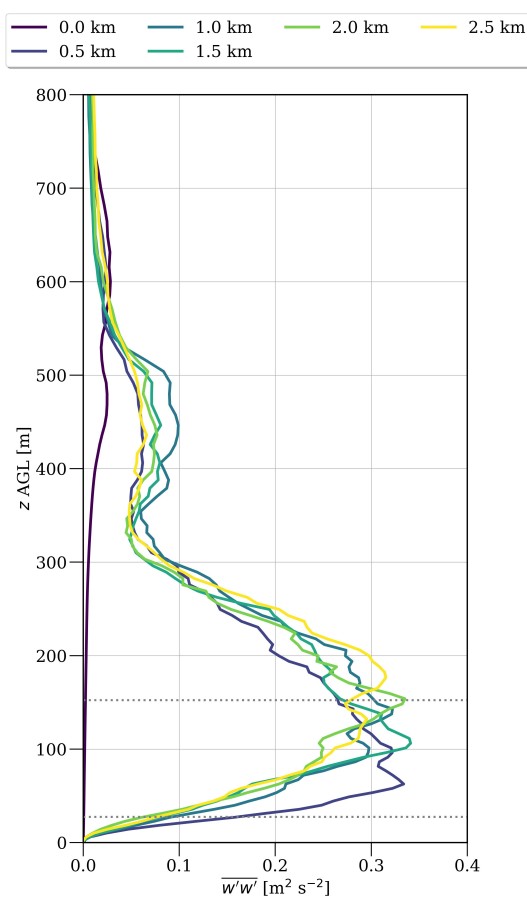

**Figure B1.** Profiles of vertical wind velocity variance ($\overline{w'w'}$) along a north-south transect at selected distances from the southern boundary inflow. The time average spans the period from 04:50 to 05:25 UTC, with a temporal resolution of 5 s.

Variability is also observed in the thrust force (Fig. C1a) and thrust coefficient ($C_t$) (Fig. C1b). These differences, combined with variations in the mean wind speed that transports wakes downstream, result in complex wind speed and wake fields. This

complexity poses challenges for effective wind farm control.

*Author contributions.* WR, BSC, JKL and NH conceptualized the work. WR carried out the simulations. WR, SL and AA did the data

curation and formal analysis. WR did the investigation. WR, ASW, MSG and RKR worked on the simulation methodology. WR prepared

the original draft. All authors reviewed and edited the manuscript.

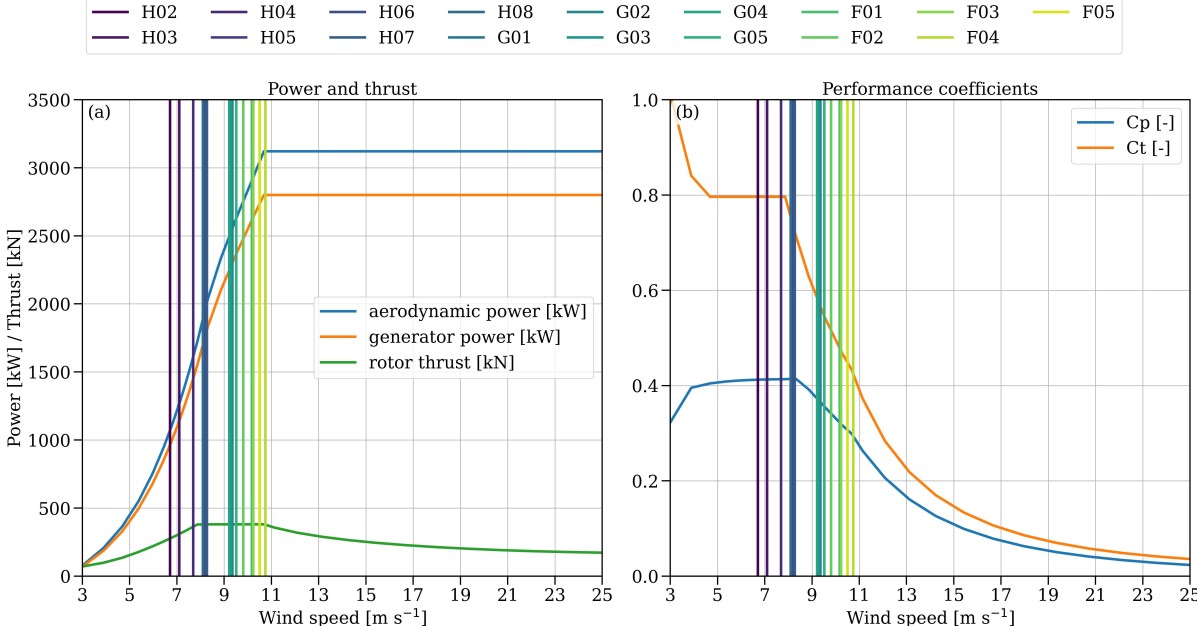

**Figure C1.** Aerodynamic, generator power, and rotor thrust curves for the NREL 2.82 MW turbine (Quon, 2022) used in the simulations (a). The power ($C_p$) and thrust ($C_t$) coefficient curves (b). The mean wind speed at hub height upstream of the turbines is calculated for the simulation with turbines and is shown as vertical lines.

*Acknowledgements.* W. Radünz would like to acknowledge the São Paulo Research Foundation (FAPESP), grant numbers 2022/04474-6
and 2023/12599-6, for financial support, and the National Renewable Energy Laboratory (NREL), operated by Alliance for Sustainable
Energy, LLC, for the U.S. Department of Energy (DOE) under Contract No. DE-AC36-08GO28308, for the Eagle supercomputer resources
necessary for the WRF simulations. Also, to Julie K. Lundquist for hosting him at the University of Colorado Boulder under the FAPESP grant
2023/12599-6. B. S. Carmo gratefully acknowledges partial support of the Coordenação de Aperfeiçoamento de Pessoal de Nível Superior -
Brazil (CAPES) - Finance Code 001, as well as support of the RCGI – Research Centre for Gas Innovation, hosted by the University of São
Paulo (USP) and sponsored by FAPESP – São Paulo Research Foundation (2020/15230-5) and TotalEnergies, and the strategic importance
of the support given by ANP (Brazil's National Oil, Natural Gas and Biofuels Agency) through the R&D levy regulation. B. S. Carmo thanks
the Brazilian National Council for Scientific and Technological Development (CNPq) for financial support in the form of a productivity
grant, number 314221/2021-2. P. Peixoto acknowledges support from the São Paulo Research Foundation (FAPESP), Grant 2021/06176-0,
and the Brazilian National Council for Scientific and Technological Development (CNPq), Grant 303436/2022-0. ASW's contributions were
prepared by Lawrence Livermore National Laboratory under Contract DE-AC52-07NA27344, with support from the U.S. DOE Office of
Energy Efficiency and Renewable Energy Wind Energy Technologies Office.





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
