# Peer review of "Influence of simple terrain on the spatial variability of a low-level jet and wind farm performance in the AWAKEN field campaign"

_Wind Energy Science, 2024_

## Referee Comment (RC2)

**Title: Influence of simple terrain on the spatial variability of a low-level jet and wind farm performance in the AWAKEN field campaign**
**Author(s): William Radünz et al.**
**MS No.: wes-2024-166**
**MS type: Research article**

This paper focuses on a strong low-level jet (LLJ) event under stable conditions during the AWAKEN campaign, investigating how simple terrain modulates flow and wake fields. Specifically, it examines how terrain-induced effects contribute to spatial variability in the flow field. For the simulation, the authors employ a multiscale modeling framework (WRF-LES-GAD) alongside available SCADA data. The study presents interesting findings with a valuable contribution to the field. Publication is recommended after addressing the comments provided for further clarity and accuracy.

**General comments**

I appreciate the authors' effort in integrating several interesting concepts into the manuscript. Given the space constraints, some of my comments stem from the manuscript's structure and the concepts addressed. For instance, while the study provides insights into the specific LLJ event analyzed, it does not sufficiently elaborate on its generation mechanism.

The paper presents several interesting findings and interpretations, particularly regarding the cross-sectional plot of KHI in Fig. 7, which effectively illustrates the flow field. However, I was expecting further discussion or analysis on how the observed instability may or may not influence the flow field or more specifically wind streamlines both above and within the wind park. Given that such instabilities can locally impact turbulence, wake dynamics, and momentum transfer, it would be important to explore whether KHI explicitly or implicitly plays a role in shaping the flow field. Including this discussion could strengthen the interpretation of the results and enhance their relevance to wind farm performance.

There are several interesting aspects in the discussion section. However, I think some parts of the Discussion section (e.g., Sections 4.2) tend to summarize previous studies in a way that resembles a literature review to me. While providing context is valuable, the main focus should be on interpreting the study's findings in a more quantitative way if it fits and highlighting their contribution to new knowledge. For example, in Sec. 4.2, the paras related to control, TI, and TKE, the authors could provide a more in-depth analysis of their own results, discussing their implications and significance in more detail and quantitative way. Strengthening this aspect

would improve the clarity of the study's contributions in order to remain centered on the novel aspects of the research.

Given the quality of the work and its good coverage of important factors, like including operational aspects such as control and aerodynamic load (e.g., in terms of Ct), I suggest that the authors explicitly discuss the effect of positive and negative shear on aerodynamic load. A brief elaboration in the main text or a relevant citation would enhance the clarity and completeness of the discussion. Additionally, I have come across a couple of reports and publications in recent years that explore multiscale interactions from mesoscale to microscale, down to structural responses, that can be cited for this purpose.

**Specific comments**

- In line 198-200, authors mention using the Eckert number (Ec) in the calculation of potential temperature perturbation amplitude, based on Muñoz-Esparza and Kosovic (2018), with a value of 0.2. It would be helpful if the authors could include and explain why the Eckert number is important in this context and how it influences the simulation, particularly in relation to the vertical confinement of perturbations and the diagnosed PBL height in this paper. Whether the 0.2 limit for the Eckert number remains the same across different applications, particularly in scenarios involving terrain effects. Does the presence of terrain influence the chosen value for the Eckert number, or is it considered a constant in all cases?
- How does your model avoid feedback between the turbulent signal in the buffer zone and the inflow boundary for the terrain simulation? More specifically, in your simulations with realistic land surface distributions, as well as simple terrain, does the method rely on statistically homogeneous turbulence within the buffer zone? If so, how can it be ensured that statistically homogeneous turbulence is achievable when large buffer zones are added?
- Aligned with above, for the innermost domain, to shorten the fetch required for turbulent spin-up (lines 196-198), cell perturbation has been used. However, the effects of this method are not limited to the potential temperature or velocity fields; it may also induce unrealistic thermodynamic conditions. This highlights the importance of having sufficient buffer zones at the inflow boundaries, where turbulence can develop spatially. As noted by Mirocha et al. (2014), who showed that without perturbations, a fetch length of several tens of kilometers is needed to achieve fully developed turbulence, meaning that a significant portion of computational resources is spent on these buffer zones. You mention in lines 202-203 that spinning up occurs between 1.5 and 2 km from the southerly boundary of D3. However, I believe more detail is needed on the

turbulence recycling process, particularly how it contributes to computational efficiency and results in faster spin-up.

- Given that the Kelvin-Helmholtz Instabilities (KHIs) are observed near the third row and below the LLJ nose, and are not caused by the turbines (as they also occur in the simulation without turbines), could the authors elaborate on the specific factors contributing to the formation of KHIs in this region (e.g. small Richardson number, …)? Are these instabilities primarily driven by shear in the wind profiles? You may slightly elaborate here with citation of any related reference on this process, for the region.

- In Figure 7 and Section 3.4, I am curious about the presence of Kelvin-Helmholtz instability (KHI) in Figures 7b and 7d and whether such instability (dynamical instability) could potentially influence streamline vertical displacement or have any potential impact on the overall flow field.

- In lines 327–336, a clearer and more quantitative explanation would be beneficial. The discussion relies on Figures C1a and C1b to explain thrust coefficient behavior, but it is somewhat unclear whether these figures present direct empirical data, simulation results, or theoretical curves. Could you clarify this? Additionally, including specific Ct values or a comparative table would enhance clarity.

  Since you are using an ADM, I assume you may have access to along-the-blade thrust force distributions or at least the total thrust force from the model. I suggest more quantitative details on this and refine the explanation accordingly.

- In Figure 9, analyzing turbulence intensity, alongside the given shear and veer studies by authors, can provide a better understanding of how terrain influences wake behavior in the wind park, as well as the relationship between TI and wake characteristics and recovery. I recommend authors could comment and elaborate on this.

**Technical corrections and minor comments**

- In the caption of Fig. 2, it appears that the domains are related to WRF, but clarifying whether they are mesoscale or microscale would be helpful. For example, the inner domain, as mentioned in lines 152 and 162, corresponds to the LES domain.

- In lines 172-173, the authors use the term 'simulations forced...'. While it is somewhat clear that the forcing files are from ERA5, the word 'forced' may give the impression of data assimilation, at least for me. While this is not necessarily incorrect, it would be helpful if the authors could slightly modify here to avoid potential confusion.

- Minor comment: The paper discusses the effect of terrain on wind speed but does not specify properly the extent of lateral inhomogeneity required to observe significant changes. A quantitative measure of inhomogeneity (e.g., spatial correlation metrics) in

the wind field during the study events would help clarify this aspect. This is particularly important given the discussion on terrain-induced accelerations, "even with simple topographic features, potentially causing substantial changes in wind speed and wind farm performance" (see lines 74–75). Additionally, it directly relates to the first study objective outlined in lines 86–87.

- Minor comment: While selecting a stationary window—where wind speed and large-scale forcing remain relatively steady—might be reasonable, could you clarify which dynamic events are relevant to this region? For example, are you referring to large-scale weather systems such as cold fronts, synoptic-scale cyclones, or other mesoscale influences? Lines 127-128.

- I may have overlooked something in Section 3.3—could you clarify what resolution is required to realistically simulate KH waves in WRF? (whether In all domains, WRF is able to resolve it ?, I assume no)? Additionally, since potential temperature serves as a useful metric for identifying wave overturning, I'm curious about how KHI responds to the selection of microphysics schemes. While this might extend beyond the immediate scope of your study, it would still be valuable if you could reference any relevant studies on KH waves in this context and area.

- Minor comment: Furthermore, is there a way to improve the representation of Figure C1? Perhaps an alternative visualization or additional annotations?.

- Minor comment: In lines 345-346 for the sake of more clarity, do you mean something like? : The initial terrain-induced up- and downward motions trigger disturbances, and the buoyancy restoring forces sustain the resulting undulations in the flow.

---

## Author Comment (AC1)

**Response to reviewers' comments**

**on the paper "Influence of simple terrain on the spatial variability of a low-level jet and wind farm performance in the AWAKEN field campaign"**

**submitted to Wind Energy Science**

Submission due on April 4 2025

Regular font – Reviewers Bold – Our response to reviewers

**RC1: 'Comment on wes-2024-166', Anonymous Referee #1, 19 Jan 2025**

**GC1 In this article, the authors investigated the impact of 'simple terrain' (characterized by slight elevation variations of less than 50 m) on wind farm power production under stable stratification with Low-Level jets (LLJs). The study utilized WRF-LES-GAD simulations and corroborated with the observational data from wind farms at the AWAKEN site. The article is well-written and the addressed details on the impact of simple terrain on wind farm performance is beneficial to wind energy community. I recommend for the publication of this article by addressing my minor comments on technical details as follows.**

**We appreciate the positive feedback from the reviewer and will do our best to address the recommendations.**

**GC2 It is unclear how the frictional velocity was evaluated for the chosen site based on the requirements outlined in lines 134-136 to determine the necessary grid resolution.**

Even though the requirement was a friction velocity lower than 0.5 m/s, the heat flux being smaller than -20 W/m2 ( $\sim$  -57 W/m2) suggests strong turbulence for nighttime. We imposed the upper limit to the friction velocity because, above that threshold, the hub height winds speeds tend to be much larger than 10 m/s, and the turbines in the first row operate near, at, or above rated capacity. That would make it difficult to evaluate the spatial variability in performance and the wakes. With that in mind, we selected the case with the highest friction velocity (~0.35–0.40 m/s, a relatively high value for nighttime hub height wind speeds of about 8 m/s during the nighttime) among the three nights that passed all the filters. This date combines relatively strong turbulence with not-too-strong hub height wind speeds, which is excellent from the point of view of our targeted research scope. Regarding the spatial resolution being sufficient for this level of turbulence, Sanchez Gomez et al. (2022) (see their Appendix) carried out a grid sensitivity study at the same site with WRF-LES-GAD for a weaker LLJ (maximum nose speed of about 21 m/s, in comparison with 25 m/s in our study) and found that a horizontal grid resolution of about 4 m was sufficient to resolve most of the turbulence above 30 m AGL. Comparing the turbulence levels in their case to ours, the observed hub height vertical velocity variance was above 0.15 m/s in their study, whereas it was 0.21 m/s in ours (about 33% larger). Thus, we conclude we can resolve most of the turbulence with a 5 m resolution grid, as further demonstrated by our agreement with observations.

We incorporated a more detailed explanation for our grid resolution choice and a citation to Sanchez Gomez et al. (2022) in lines 159-165, as below:

"The innermost nest domain (D3) has a fixed horizontal resolution of \$\Delta x=\$~5~m, determined based on a grid sensitivity study conducted at the same site using WRF-LES-GAD for a weaker LLJ case (see the Appendix of Sanchez Gomez et al. (2022)). Their study shows that most turbulence above 30 m AGL is adequately resolved

**with a 3.94 m grid. In our LLJ case, turbulence is stronger, with a vertical velocity variance of approximately 0.21 m2/s2 at 90 m AGL, compared to 0.15 m2/s2 in their study. To ensure turbulence is adequately resolved, we use a 5 m grid in the innermost domain."**

**GC3 Details regarding the simulation setup for domains D1, D2, and D3 are not sufficiently clear, particularly how the simulations were conducted. Including a flowchart of the simulation setup, either in the Appendix or Section 2.2, would help clarify this. Such a flowchart could detail the initial conditions based on the HRRR model and their integration with the GAD framework, which would be beneficial for readers.**

We have included a flowchart of the simulation framework that is focused on the communication between the input data (HRRR) and the WRF domains via initial and lateral boundary conditions (I/LBCs) in Appendix D. It also briefly describes how the CPM and the GAD integrate the multiscale framework.

Figure D1 – Multiscale simulation flowchart: The HRRR analysis dataset (hourly frequency) provides initial and lateral boundary conditions (I/LBCs) for the mesoscale domain D1 in WRF. The nested LES domains D2 and D3 receive I/LBCs from their respective parent domains (D1 and D2) through one-way coupling, meaning no feedback occurs to the parent domains. To reduce computational cost, the domains are activated sequentially in time. The Cell Perturbation Method (CPM) and Generalized Actuator Disk (GAD) are applied only in the innermost domain D3.

**GC4 What do the black vertical lines in Figures 11 and 12 represent? Additionally, what is the significance of the dotted lines in Figure 4? I would expect them to indicate the tips of the wind turbine blades; however, please confirm this and make any necessary changes to the figure captions.**

The black vertical lines in Figures 11 and 12 are the projected positions of the H05, G02, and F04 turbine rotors onto the vertical plane. Also, the reviewer is correct about Figure 4. The dotted horizontal lines represent the rotor top and bottom tips of the turbines. Finally, we have adjusted the captions of Figures 4, 11 and 12, as below:

Figure 4. Vertical profiles of wind speed (a), direction (b), potential temperature (c), TI (d) and w'w' (e) for a 30 minute window between 04:55 and 05:25 UTC. Observations from the scanning lidar at site A1 (OBS-A1-SL), the profiling lidar at site A1 (OBS-A1-PL), and the AERI at site C1 (OBS-C1-AERI) are represented as markers. Results from domain D3 are represented as blue continuous lines. The dotted horizontal lines represent the rotor top and bottom tips of the turbines.

Figure 11 – Figure 11. Vertical cross-section of wind speed difference relative to the front row profile (WS – WS fr) for domain D3 in the simulation without turbines. The LLJ nose height is also displayed (a). Wind speed difference along terrain-following lines at fixed heights AGL (b). The black vertical lines indicate the projected positions of turbine rotors H05, G02, and F04 onto the vertical plane.

Figure 12 – Vertical cross-section of wind speed difference relative to the front row profile (W S – W Sfr) for domain D3 in the simulation without turbines, with streamlines of the mean flow located between 20 and 600 m AGL near the inlet (a). The remaining subplots show the vertical displacement along the streamlines ( $\Delta$ z, b), actual change in wind speed along the streamlines ( $\Delta$ WS str, c), estimated change in wind speed caused by the vertical displacement of the LLJ ( $\Delta$ WS vert, d) and the estimated total change in wind speed of (c) and (d) combined ( $\Delta$ WStot, e). The aforementioned variables are relative to the value at the front row The black vertical lines indicate the projected positions of turbine rotors H05, G02, and F04 onto the vertical plane.

**GC5 The variables used in the text are typeset with LaTeX, whereas those in the figures are not. While this is a minor point, maintaining consistency between the text and figures would enhance the overall presentation.**

We appreciate the reviewer's attention to detail, and we agree that having the Figures with variables consistently typeset improves the manuscript. Thus, we have modified all the Figures with the same rule such as shown in Figure 3.

---

## Author Response (AR2)

**Response to the Editor's Comments**

Manuscript title: Influence of simple terrain on the spatial variability of a low-level jet and wind farm performance in the AWAKEN field campaign

Journal: Wind Energy Science

Editor's comments – Regular font Authors' comments – **Bold font**

Public justification (visible to the public if the article is accepted and published):

Dear authors.

The two reviewers are satisfied with the revisions to the manuscript and recommend publication as it is. I agree with their assessment, but I have a few minor corrections listed below. Thank you for choosing WES for your publication, and once again, I apologize for the lengthy review process.

**Dear Editor,**

We thank you for your careful reading and helpful suggestions. Please find our point-by-point responses below:

L67-68. "power performance" should perhaps be "the performance of a wind turbine" or something like this. Wind turbines suddenly appear in the text

The phrase "power performance" was revised to clarify the context and introduce wind turbines earlier. The sentence now reads: "The terrain-induced variability in wind speed during SBLs and LLJs causes important spatial differences in the performance of wind turbines operating in complex terrain."

L70-71. "wind farms built in complex." Something about wind farms is being studied; otherwise, the sentence is a bit weird.

The sentence was revised for clarity. It now reads: "In Radünz et al. (2021), wind farms located in complex terrain exhibited surprising performance

patterns: turbines in the back rows sometimes produced twice as much power as those in the front rows, despite being affected by wake effects."

L80. IMO "American WAKe ExperimeNt" is not necessary, just write "American Wake Experiment"

We appreciate the suggestion regarding the stylization of "American WAKE ExperimeNt." However, we chose to retain the stylized version to remain consistent with the official naming convention used by the AWAKEN project consortium, as published in Moriarty et al. (2024) and Bodini et al. (2024). The full name appears only twice in the manuscript, and we otherwise refer to the project as AWAKEN throughout.

Moriarty et al. (2024). Overview of preparation for the American WAKE ExperimeNt (AWAKEN). *Journal of Renewable and Sustainable Energy*, *16*(5). https://doi.org/10.1063/5.0141683

Bodini et al. (2024). An international benchmark for wind plant wakes from the American WAKE ExperimeNt (AWAKEN). *Journal of Physics: Conference Series*, 2767(9). https://doi.org/10.1088/1742-6596/2767/9/092034

In several places: I would add "model" after WRF; otherwise, the sentences don't make sense.

We have added the word "model" after "WRF" in all relevant instances to improve clarity.

Figures 10-12. If it is easy to do, could all the x-axis be of identical size? I think it will facilitate understanding.

We revised these figures so that all x-axes now have identical sizes, as suggested.

Please update the reference to Wise et al. 2024; the paper has now been published. Please ensure all citations are valid.

The reference to Wise et al. (2024) has been updated. All citations have been verified for consistency, and journal names were abbreviated accordingly.

We also appreciate your kind acknowledgment regarding the length of the review process and fully understand that such delays can happen. Thank you again for your guidance and for the opportunity to publish our work in Wind Energy Science.

Sincerely, The authors